# Brightness illusions drive a neuronal response in the primary visual cortex under top-down modulation

Alireza Saeedi [1,10], Kun Wang[1,2], Ghazaleh Nikpourian[1], Andreas Bartels [1,3,4], Nikos K. Logothetis[1,2,5], Nelson K. Totah [1,6,7,8] ✉ & Masataka Watanabe [1,9] ✉

Brightness illusions are a powerful tool in studying vision, yet their neural correlates are poorly understood. Based on a human paradigm, we presented illusory drifting gratings to mice. Primary visual cortex (V1) neurons responded to illusory gratings, matching their direction selectivity for real gratings, and they tracked the spatial phase offset between illusory and real gratings. Illusion responses were delayed compared to real gratings, in line with the theory that processing illusions requires feedback from higher visual areas (HVAs). We provide support for this theory by showing a reduced V1 response to illusions, but not real gratings, following HVAs optogenetic inhibition. Finally, we used the pupil response (PR) as an indirect perceptual report and showed that the mouse PR matches the human PR to perceived luminance changes. Our findings resolve debates over whether V1 neurons are involved in processing illusions and highlight the involvement of feedback from HVAs.

Non-human primate and cat V1 single cells respond to physical brightness[1–6], but whether V1 single cells respond to illusory brightness is controversial[7,8]. For instance, it has been reported that V1 neurons in monkeys respond to real surface luminance, but not to illusory brightness[7], and one study in cats reported single neuron responses to illusory brightness in both V1 and V2 but found much higher prevalence in V2[8]. Moreover, fMRI studies in humans have been inconclusive, with some studies reporting a correlation between V1 BOLD signal and perception of illusory brightness[9–11], while others found no such correlation[12–14]. Thus, the V1 neuronal response to illusory brightness is uncertain in humans and animals[15–18].

In the present study, using in vivo electrophysiology in mouse V1, we probe the neural correlates of an achromatic version of neon color spreading (NCS)[19,20], an illusion that has been shown in humans to combine different perceptual qualities, such as filling-in and perception of illusory contours and brightness[21]. We show that single units recorded in mouse V1 respond to NCS stimuli designed to generate an illusory drifting grating. Neuronal responses were compared with a physical drifting grating in antiphase relative to the illusory grating. Analyzing the spatial phase tuning properties of single units allowed us to demonstrate that V1 single units respond to illusory brightness as though a real grating was presented.

Using this mouse paradigm, we were able to probe the putative neuronal organization of the V1 microcircuit[22–26] involved in illusory brightness processing by studying the relationship between single unit responses to the illusory stimuli and surround modulation[27,28], complex-simple cell modulation[29–32], as well as by characterizing units as putative inhibitory interneurons and putative excitatory

[1]Department of Physiology of Cognitive Processes, Max Planck Institute for Biological Cybernetics, 72076 Tübingen, Germany. [2]Department of Physiology of Cognitive Processes, International Center for Primate Brain Research, Songjiang District, Shanghai 201602, China. [3]Department of Psychology, Vision and Cognition Lab, Centre for Integrative Neuroscience, University of Tübingen, Tübingen, Germany. [4]Bernstein Center for Computational Neuroscience, Tübingen, Germany. [5]Centre for Imaging Sciences, University of Manchester, Manchester M139PT, UK. [6]Helsinki Institute of Life Science (HILIFE), University of Helsinki, 00014 Helsinki, Finland. [7]Faculty of Pharmacy, University of Helsinki, 00014 Helsinki, Finland. [8]Neuroscience Center, University of Helsinki, 00014 Helsinki, Finland. [9]Department of Systems Innovation, School of Engineering, The University of Tokyo, Tokyo, Japan. [10]Present address: Research Group Neurobiology of Magnetoreception, Max Planck Institute for Neurobiology of Behavior – caesar, 53175 Bonn, Germany. ✉e-mail: nelson.totah@helsinki.fi; watanabe@sys.t.u-tokyo.ac.jp

pyramidal neurons[33,34]. A prominent theoretical view is that feedback from higher visual areas (HVAs) is necessary to process illusions[35,36]. We tested this using optogenetic inhibition of HVAs and showed that disrupting their activity suppresses the V1 single unit response to the illusory brightness.

A fundamental challenge in studying visual illusions in animal models is determining whether animals can"perceive" illusions without a language-based system for reporting subjective experiences. Pupillometry has been used to infer perceptual experiences of brightness illusions in humans, as well as in rats, and is a promising method to infer perceptual experience in pre-linguistic humans (e.g., babies)[37–41]. Here, we used pupillometry in mice as an indirect behavioral response of perception. We found that the illusory brightness evokes a pupillary response in mice indicative of illusory brightness perception.

## Results

We performed in vivo electrophysiology on 19 awake, head-fixed mice passively viewing visual stimuli (Fig. 1). The mice were positioned on a disk and free to run or remain immobile. Before the experiment, we made two preliminary recordings aimed at locating single unit receptive fields (RFs) and characterizing size tuning. Briefly, we first performed RF mapping to estimate the center of the RF using the multi-unit response to black rectangles (covering 15° of the visual field) presented on a gray background at locations selected in a pseudo-random order from an 8 by 13 grid. Next, we recorded unit responses to circular patches of drifting gratings with different sizes (2.5°–45° coverage of the visual field) and presented them in the center of the RF. These recordings were subsequently used to characterize size tuning. Following these preliminary recordings, we began recording the neuronal responses to the stimuli shown in Fig. 1 to study whether V1 units respond to illusory brightness.

We presented three types of stimuli, either with a full-screen size (covering 130.7° of the visual field) or centered on the RF (covering 35° of the visual field). The first stimulus type was an achromatic version of the NCS stimulus[21] (Fig. 1a and Supplementary Fig. 1a). This stimulus consisted of an array of white concentric circles presented on a black background. Each of the concentric circles in the array contains gray segments at different positions. These gray segments are aligned in a way that diffusion of gray color into the background produces an illusory grating that seems darker than the surrounding black background. Depending on which segments of the concentric circles are gray, the orientation of the illusory grating changes. Gray segments were introduced at distinct positions in each time frame, which produced an illusory grating that drifts (Supplementary Movie 1).

The second and third types of stimuli were control conditions. The diffusion-blocked control (DBC) stimulus (Fig. 1c and Supplementary Fig. 1b) had identical temporal dynamics to the NCS stimulus. However, each concentric circle was constrained by two static circles, which led to the extinction of the illusory grating by disrupting the diffusion of brightness (Supplementary Movie 2). The DBC stimulus was used to verify that a V1 response to the NCS stimulus was due to the processing of an illusory grating, as opposed to the physical stimulus changes within the RF. The other control stimulus was a luminance-defined grating (LDG), which was a real drifting grating presented in the foreground over the concentric circles (Fig. 1d, Supplementary Fig. 1c, and Supplementary Movie 3). The LDG stimulus had the same spatial and temporal frequency as the illusory grating generated by the NCS stimulus. It served as a control condition in which a physical drifting grating was presented in order to compare the tuning properties of units for illusory gratings (NCS stimuli) with real gratings (LDG stimuli). The same gray color was used in all conditions to have comparable neuronal responses. For all three stimulus types, we presented eight drift directions that were selected in pseudo-

random order. The presentation of NCS, DBC, and LDG stimuli was also in a pseudo-randomized order.

## Mouse V1 single units respond to NCS stimuli

We analyzed the stimulus-evoked spiking of 520 mouse V1 single units ($N = 6$ mice). Example responses evoked by the NCS, LDG, and DBC stimuli are shown in Fig. 2a, b. Additional examples are presented in Supplementary Fig. 2. We found that 57.2% of the units responded to both NCS and LDG stimuli, 39.5% responded to only LDG stimuli, and 3.3% responded to only NCS stimuli (Fig. 2c). The magnitude of the response to NCS stimuli was significantly smaller than the response to LDG stimuli (linear mixed-effects model (LMEM): $F = 29.34$, $p = 7.53e\text{-}8$; Fig. 2d). In contrast to the LDG and NCS stimuli, the DBC stimuli did not evoke any response despite its exact pixel-wise changes compared to the NCS stimulus (Fig. 2c, e, f). Importantly, the lack of response to the DBC stimuli demonstrates that the NCS response is not due to the local physical changes of stimulus. Moreover, the lack of response to the DBC stimulus is consistent with the illusion being abolished so that no grating (even illusory) was present and able to drive a V1 neuronal response. Thus, V1 single units respond to the NCS stimulus but not when the illusory grating is extinguished in the DBC stimuli.

However, it has been shown that locomotion modulates the stimulus-evoked response of visual cortex neurons[42,43] and it is possible that the NCS response could be due to locomotion. We assessed how running speed affected the V1 unit responses to LDG and NCS stimuli. Consistent with prior studies[42,43], stimuli presented during running (i.e., mean running speed of >1 cm/s in a 500 ms window starting 300 ms prior to stimulus onset) evoked a larger response to both NCS and LDG stimuli (Supplementary Fig. 3a, b). We defined an illusory grating response (IGR) index that quantifies the preference of units for the NCS stimuli relative to the LDG stimuli. A positive IGR indicates a larger evoked response to the NCS stimulus, whereas a negative value indicates a larger LDG-evoked response (see Methods section, Eq. 1). Importantly, we observed that the IGR index did not change in trials with running compared to those without running (LMEM: $F = 2.26$, $p = 0.13$; Supplementary Fig. 3c). It can, therefore, be ruled out that running altered the preferred stimulus of each unit or that it differentially modulated responses to the LDG and NCS stimuli. The V1 unit response to illusory brightness is not due to locomotion.

Since V1 neurons exhibit preferred angles for drifting gratings across species, including mice[44–46], we reasoned that if mouse V1 units respond to illusory gratings as though they are like gratings that are physically present, then each unit's preferred angle would be similar for real and illusory gratings. We obtained the preferred angle of each unit using the eight drift directions of the LDG and NCS stimuli. The preferred angle was defined as the drift direction that evoked the maximal response for each unit. We found that the preferred angle was invariant for most units when comparing the LDG and NCS stimuli (Fig. 2g–i). Therefore, for any given preferred angle determined by real gratings, the unit tended to prefer the same angle when illusory gratings were evoked by NCS stimuli.

Given that humans perceive the illusory grating as darker than the surrounding black background (Fig. 1a, Supplementary Movie 1), we presented NCS and LDG stimuli with a 180° relative luminance phase shift (Fig. 2j) in order to demonstrate that mouse V1 units respond to the illusory grating as if the bars are darker than the surrounding black background. Such a result would strongly support that mouse V1 units respond to illusory brightness. In order to demonstrate this property in V1 units, we tested the hypothesis that unit responses preserved the spatial phase properties of the stimuli. As shown in rasters and peri-stimulus time histograms (PSTHs) of example units, the response to NCS stimuli was shifted compared to the response to LDG stimuli (Fig. 2a, b). We quantified this effect by calculating the phase shift between the first harmonic

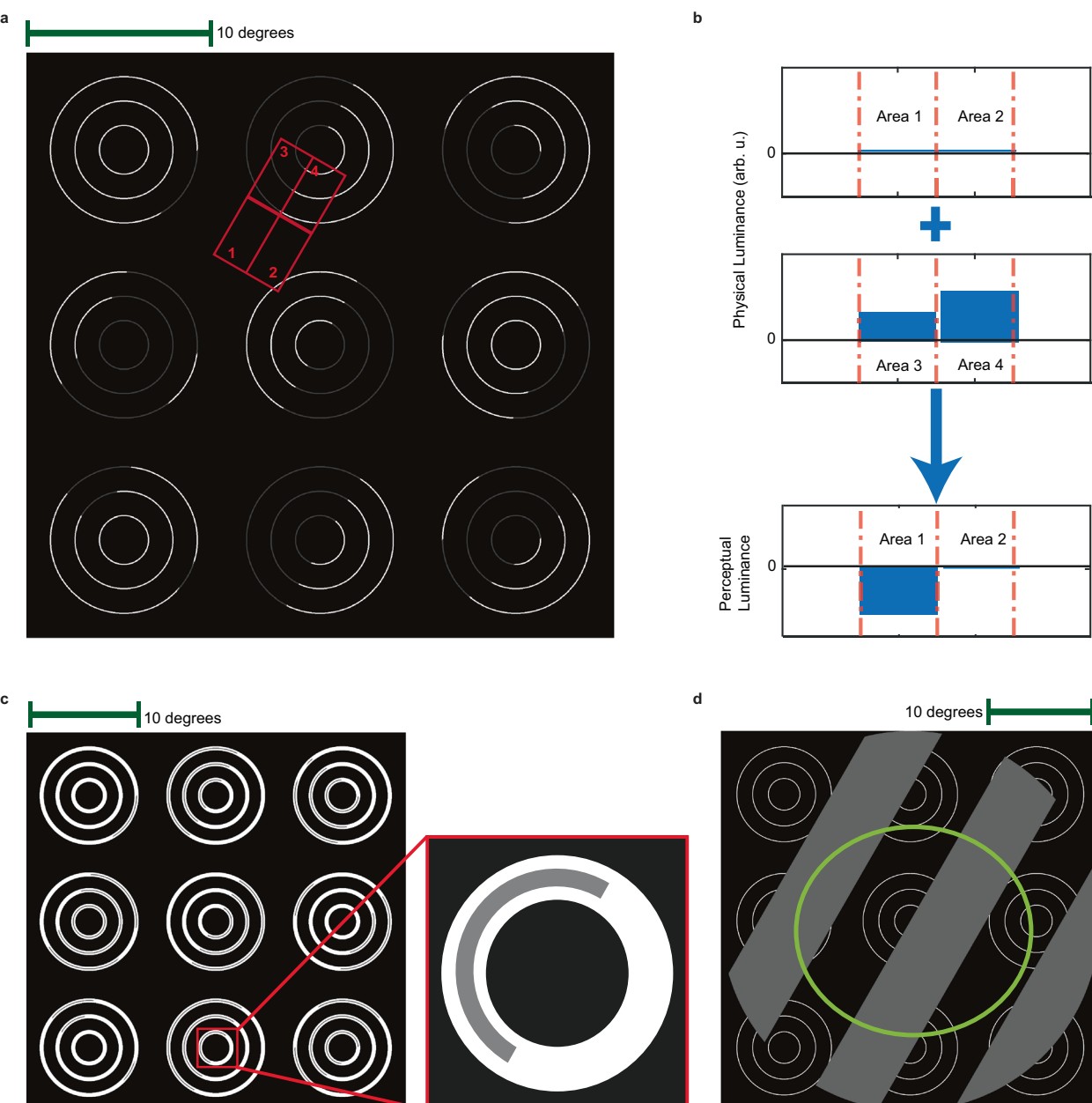

**Fig. 1 | The visual stimuli consisted of NCS stimuli that evoked illusory luminance to form a drifting grating and control stimuli that either blocked the illusion (DBC stimuli) or were a real grating (LDG stimuli). a** An example of the achromatic NCS stimulus presented to mice. The stimulus consists of 9 sets of white concentric circles. Each set of concentric circles contains gray segments at different positions. As can be seen, the gray color is diffusing into the empty area between concentric circles. This generates an illusory grating that appears darker than the surrounding black background. Changing the location of the gray segments on the concentric circles will change the orientation of this illusory grating. **b** The schematic shows the physical and perceptual luminance in the four arbitrary areas bounded by red boxes and marked as areas 1, 2, 3, and 4 in panel a. Luminance differences in areas 3 and 4 make an illusory perceptual luminance difference between areas 1 and 2 due to the diffusion of the gray color from area 3 to area 1. The luminance values are arbitrary. **c** In the DBC stimulus condition, the concentric circles are constrained by an outer white"band," which serves to block the diffusion of gray color while maintaining the presentation of the physical stimulus (i.e., the concentric circles). **d** A control stimulus that provided a real luminance-defined grating (LDG) was used to compare an actual grating with the illusory grating evoked by the NCS illusion in panel a. This grating was presented over a background compound of a steady concentric circle with a temporal and spatial frequency identical to the illusory grating generated by the NCS stimulus. The green ellipse shows an example of a V1 RF for comparing its size relative to the stimuli.

(F1 component) of the neuronal responses to NCS and LDG stimuli. In order to ensure the reliability of the calculated phase, this analysis was only applied to a subset of units ($N = 209$) in which their F1 component was the dominant frequency component (i.e., the power of the F1 component corresponding to the 2 Hz temporal frequency of the grating was larger than all other non-zero components). An example of an F1 dominant unit is shown in Fig. 2k. We found that the phase shift between the NCS response and LDG response was significantly non-uniform (Rayleigh's test, $Z = 50$, $p = 7.3\text{e-}24$) and tightly distributed around a circular mean of 178.23° (95% confidence interval = [167.66°, 188.90°]). This anti-phase response to NCS and LDG stimuli corresponds to the anti-phase luminance perceived by humans viewing these stimuli (Fig. 2j). This correspondence indicates that mouse V1 units

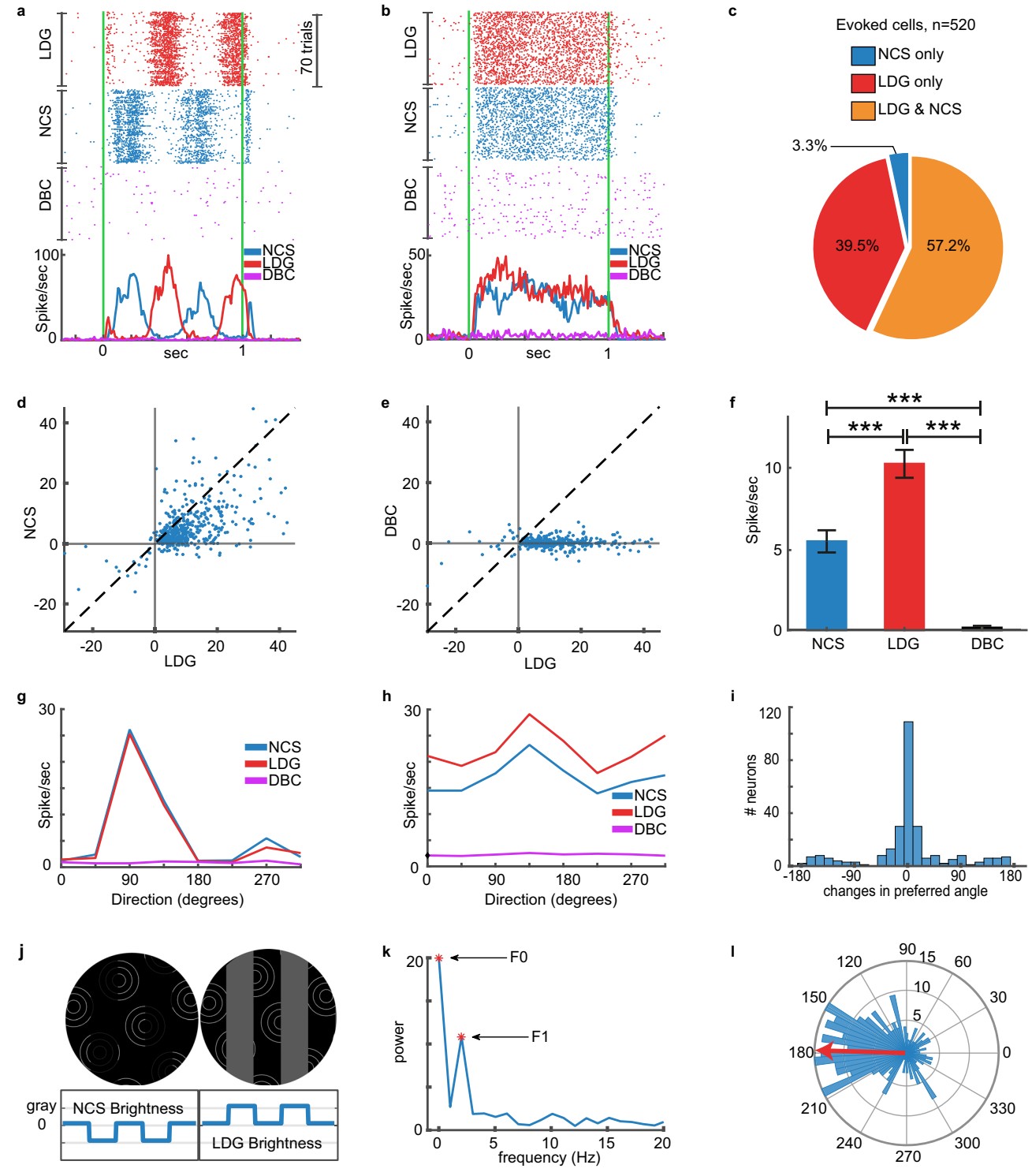

respond to illusory brightness in the form of grating bars that are darker than the surrounding black background. Overall, the results presented in Fig. 2 indicate that mouse V1 units respond to the illusory gratings evoked by NCS stimuli in a fundamentally similar manner to how they respond to real gratings.

## V1 single units respond to NCS stimuli in the absence of RF stimulation

The DBC stimulus (Fig. 1c) was designed to eliminate the filling-in effect and extinguish the illusory grating, and we showed that V1 units did not respond to the DBC stimulus (Fig. 2a, b, f). However, the response to

the NCS stimulus could be due to direct RF stimulation. We eliminated this possibility by performing a second experiment. After the RF mapping, we presented a full-screen version of the NCS, LDG, and DBC stimuli with larger distances between patches of concentric circles (Supplementary Fig. 1). The greater distance between the inducers permitted the recording of additional units without RF-inducer overlap. We inserted the electrode with an oblique (30 degrees lateral) angle to cross multiple cortical columns with a variety of RF screen locations.

In 13 mice, we recorded 1807 V1 single units, of which 234 units had an RF-inducer overlap of less than 1% of their RF size and no

**Fig. 2 | Single units respond to illusion. a**, **b** Rasters and peri-stimulus time histograms (baseline-subtracted) of two V1 units in response to physical gratings (LDG), illusory gratings (NCS), and diffusion-blocked illusory gratings (DBC) at the preferred direction of each unit (90° and 135°, respectively). Green lines indicate stimulus onset and offset. **c** The pie chart shows the percentage of single units responding to various stimuli. **d** The scatter plot illustrates the maximal (trial-averaged) firing rate (spike/sec) of all 520 visually responsive units for NCS stimuli against LDG stimuli at the preferred orientation of each unit. **e** The scatter plot shows the maximal response of all 520 units for LDG and DBC stimuli. Plotting conventions are identical to (**d**). **f** The bar plot shows the average response magnitude across all 3 stimulus types (*N* = 520 units). Error bars indicate 95% confidence intervals. Kruskal-Wallis test H(2) = 26665, *p* < 0.0001 and the post-hoc multi-comparison (Tukey's Honestly Significant Difference Procedure). *** indicates *p* < 0.0001. **g**, **h** These panels show the direction tuning curves of the two example

single units shown in (**a**, **b**). **i** The histogram shows the differences in the preferred angle between responses to NCS and LDG stimuli for the 297 units (57.2% of 520 units) that responded to both stimulus types. **j** Due to the location of the gray segments in the NCS stimuli, the illusory grating is 180° out of phase with the physical grating (LDG stimuli). Example stimuli and a schematic of grating bar brightness are shown to illustrate that the dark bars of the illusory grating are aligned with the light bars of the physical grating. **k** Frequency profile of single unit with an F1 dominant component of the power spectrum. Stars show the F1 and F0 components. **l** The polar plot shows the phase shift between the response evoked by NCS and LDG stimuli (*N* = 209 units with an F1 dominant component in the evoked response). The red arrow shows the angular mean of this circular distribution (178.2°). The radial numbers indicate the number of neurons in the histogram. *N* = 6 mice. Source data are provided as a Source Data file.

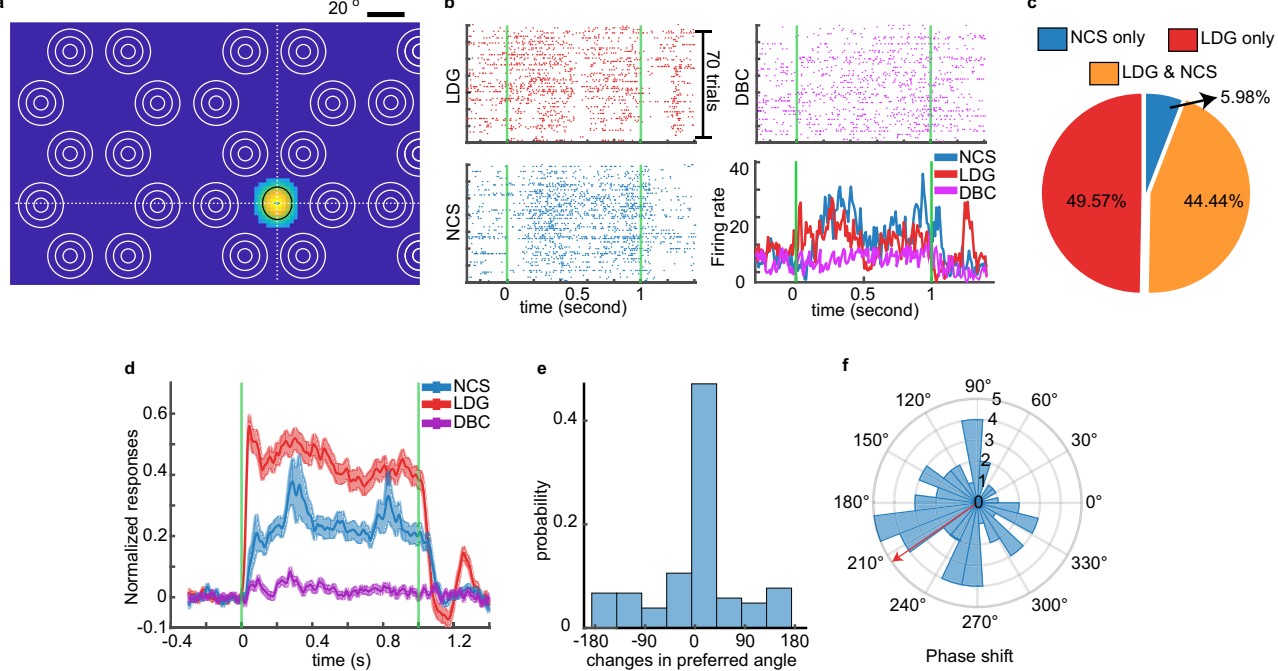

**Fig. 3 | The V1 single units response to NCS stimuli is not due to inducer overlap with the RF. a** RF of an example non-overlapping unit. The color illustrates the magnitude of the unit response to the black rectangle during the RF mapping session. The black ellipse indicates the full width at half maximum of a 2-d Gaussian function fitted to the unit response. The intersection of 2 dashed white lines is at the center of the RF. The white concentric circles used in NCS stimuli are shown for comparison with the RF location of this unit. **b** Stimulus-evoked spike rasters and PSTHs of the example unit shown in (**a**). **c** The pie chart shows the percentage of non-overlapping units that responded to each stimulus type (*n* = 234 units). **d** The

normalized population response magnitude across all 3 stimulus types for the non-overlapping units. Shading indicates standard deviation. **e** The distribution of differences in the preferred angle between responses to NCS and LDG stimuli (*n* = 104 units that responded to both stimulus types). **f** The circular distribution of the phase shift of the response evoked by the NCS stimulus relative to the LDG stimulus (*n* = 55 units with an F1 dominant component in the evoked response). The red arrow shows the angular mean (214.55°). The radial numbers indicate the number of units. *N* = 13 mice. Source data are provided as a Source Data file.

response to DBC stimulus. Units without RF-inducer overlap ("non-overlapping units") still responded to the NCS stimulus (Fig. 3a, b). This response was observed in 104 units, which is approximately half of the recorded units (Fig. 3c). At the population level, the NCS stimulus-evoked response of these 104 units was present but again reduced compared to the LDG stimulus-evoked response (Fig. 3d). There was only a weak correlation between the IGR and the degree of RF-inducer overlap (Pearson's correlation coefficient: *r* = 0.09, *p* = 6.23e-3), Supplementary Fig. 4, which means that the preference to respond to the NCS stimulus slightly increases as its RF encroaches upon the physical luminance change of the inducer. Therefore, units with an RF in the "dark" regions of the NCS stimulus still respond to the NCS stimulus. We conclude that the response to the NCS stimulus is not purely caused by physical luminance changes due to inducers in unit RFs. The non-overlapping units also

maintained their shared preferred angle for real gratings (as determined by LDG stimuli) and illusory gratings (evoked by NCS stimuli) for most of these 104 units (Fig. 3e). We also calculated the F1 phase shift between the NCS response and LDG response for the subset of units with a dominant f1 component (55 units) and found that the phase shift was significantly non-uniform (Rayleigh's test, Z = 3.52, *p* = 0.028) with a circular mean of 214.55° (95% confidence interval = [167.4°, 261.6°], Fig. 3f). The phase shift is robust as it holds for other units with varying amounts of RF-inducer overlap and becomes more closely aligned to a mean of 180° as the RF and inducer overlap to a greater extent (Supplementary Fig. 5). However, even with no overlap, there is still considerable phase shift with a confidence interval containing the 180° antiphase relationship between the NCS and LDG stimuli. We conclude that the response to the NCS stimulus is due to the illusory grating induced by the

concentric circles and not direct RF stimulation by small luminance changes driven by the overlap of the inducer with the RF of V1 single units.

## The response to illusory brightness is delayed relative to responses to real stimuli

We next examined the level within the V1 local cellular organization where neural processing of the brightness illusion occurs. Studies on humans[47], macaque[35], and mice[36] have found that the neuronal response to visual illusions is delayed compared to real stimuli, which may suggest that the neural correlates of visual illusions could be at a different level within the V1 cellular hierarchy. For instance, a later response to a stimulus might be due to additional serial synaptic interactions within V1 (and also potentially from top-down inter-cortical poly-synaptic inputs from higher levels of the cortical hierarchy that may contribute to illusory perception)[35,36,47,48]. We calculated the latency of the NCS and LDG stimulus-evoked responses using only stimuli presented at the preferred direction of each unit[49] (see Methods section for a description of how latency was calculated). We excluded simple cells, which have a phase-locked response to the drifting grating and for which it is not possible to estimate response latency. We found that NSC stimulus-evoked responses are later than responses to LDG stimuli (Fig. 4a). The latency (mean ± SEM) was 65.74 ± 0.17 ms for LDG but increased to 99.18 ± 0.33 ms for NCS stimuli (LMEM: $F = 40.13$, $p = 4.62e-10$). It should be noted that this held for units without RF-inducer overlap (mean ± SEM for NCS was 112.69 ± 7.45 compared to 62.39 ± 5.6 for LDG; LMEM: $F = 29.02$, $p = 2.59e-7$). These data suggest the potential that the neural correlates of illusory brightness in V1 require additional serial synaptic processing.

However, an alternative explanation, which excludes additional serial synaptic processing, is that the response to the LDG stimulus occurs earlier because it has more physical luminance compared to the NCS stimulus. The slower latency of the NCS stimulus could be due to its lower physical luminance activating feedforward inputs more slowly, as opposed to it requiring a greater number of serial synaptic activations. We tested this explanation by comparing response latencies when the response magnitude was matched for the NCS and LDG stimuli. We identified 286 equi-responsive complex cells for this analysis. The response to NCS stimuli remained significantly delayed compared to LDG stimulus-evoked responses (LMEM: $F = 30.74$, $p = 4.31e-8$; Supplementary Fig. 6).

We also assessed whether the processing delay could be explained by the fact that the NCS stimulus drives Off RFs, which could potentially have a delayed response relative to the On RFs activated by the LDG stimulus. The black rectangles presented during the initial RF mapping experiment provide a convenient means of characterizing the response latency of Off RFs. The latency of the neuronal response to the black rectangles used in the RF mappings is a generic measure that can be calculated for simple and complex cells because there is no phasic component in the stimulus. We termed this measure the "rectangle latency." We estimated the rectangle latency for the 381 units with an evoked response. The rectangle latency was 42.76 ± 0.75 ms (mean ± SEM). The PSTH of some example units and their rectangle latencies are depicted in Supplementary Fig. 7. We compared the NCS latency to the rectangle latency in the subset of units for which we could calculate both latencies (236 units). The NCS responses were significantly delayed compared to responses to rectangles (95.03 ± 4.67 ms versus 40.29 ± 0.94 ms for NCS or rectangle, respectively; mean ± SEM; LMEM: $F = 16.80$, $p = 4.92e-5$). A similar latency difference was observed for units with limited RF-inducer overlap (mean ± SEM: 51.97 ± 2.1 ms rectangle latency compared to 112.69 ± 7.45 NCS response latency; LMEM: $F = 9.64$, $p = 2.48e-3$). Overall, these results demonstrate that the V1 neuronal response to illusory gratings is delayed relative to real gratings. Importantly, we

eliminated at least two potential causes of the delayed response to illusory brightness in the NCS stimulus: the Off RF response to NCS stimuli and the difference in the magnitude of the evoked response between NCS and LDG stimuli. Therefore, the delay may be due to V1 neuronal processing of the illusory gratings requiring additional serial synaptic activations within the V1 microcircuit in comparison to real gratings.

If NCS responsive neurons are indeed activated at a later stage of feed-forward processing in the V1 microcircuit, then these neurons should have a later response to real stimuli in general[48]. We hypothesized that neurons with a positive IGR (i.e., a preference for the NCS stimulus relative to the LDG stimulus) would have a longer rectangle latency in line with these neurons being activated at a later stage of bottom-up serial synaptic activation in V1. We tested this hypothesis by calculating the correlation between the rectangle latency and the IGR index. Our analysis revealed that IGR was actually anti-correlated with rectangle latency (Pearson's correlation coefficient: $r = -0.31$, $p = 5.17e-10$, Fig. 4b). This result suggests that the delayed response to the NCS stimulus may not be due to additional serial synaptic activations within V1, but could instead be due to serial synaptic activations that occur outside of V1 (e.g., from higher visual areas) that may be required for the V1 neuronal response to illusory brightness.

## NCS-responsive units have different functional and physiological properties

Next, we probed the V1 microcircuit more deeply and defined what additional components of the microcircuit are engaged during illusory brightness processing compared to real gratings. We began by assessing the relationship of the role of surround modulation, as it has been suggested that this is a result of intra-V1 horizontal connections[50–53] (although, note that it also depends on inter-cortical feedback/feed-forward connections[24,26]). We hypothesized that V1 units with more robust surround modulation are preferentially responsive to NCS stimuli. We tested this hypothesis by calculating the correlation between the IGR index and a surround modulation index (see Methods section, Eq. 2). The surround modulation index was calculated using the size tuning curves of each unit. This index is negative when stimulus presentation outside the classical RF facilitates the firing rate (i.e., a so-called 'facilitative cell') and is positive when extra-classical RF stimulation suppresses the firing rate (i.e., 'suppressive cell'). Supplementary Fig. 8 shows the size tuning of example facilitative and suppressive cells. We found that IGR and the surround modulation index were positively correlated (Pearson's correlation coefficient: $r = 0.26$, $p = 4.6e-8$). The relationship between these variables is shown in Fig. 4c. This result indicates that a greater response to NCS stimuli (relative to LDG stimuli) is associated with positive surround modulation indicative of suppressive cell activation. When we separated NCS preferring units (with higher responses to NSC stimuli) and LDG preferring units (with higher responses to LDG stimuli), we found that LDG preferring units had a mean ± SEM surround modulation index of 0.22 ± 0.01, indicating weak surround modulation (Fig. 4d). On the other hand, for NCS preferring units, the mean ± SEM was 0.53 ± 0.03. The difference in surround modulation index between these unit sub-populations was significant (LMEM: Fstat. = 57.60, $p = 1.73e-13$). These results indicate that NCS preferring units have more robust surround modulation in comparison to LDG preferring units and that illusory brightness processing may involve V1 suppressive cells. These findings provide evidence supporting the notion that V1 units that respond to illusory brightness receive greater intra-V1 horizontal connections (although inter-cortical connections may also play a role).

Another property of V1 cells related to the functional organization of the V1 microcircuit is their complex versus simple cell designation[29–32]. Unlike simple cells, there are no segregated excitatory/inhibitory areas in the RF of complex cells; therefore, their responses are not locked to the phase of the drifting grating. The

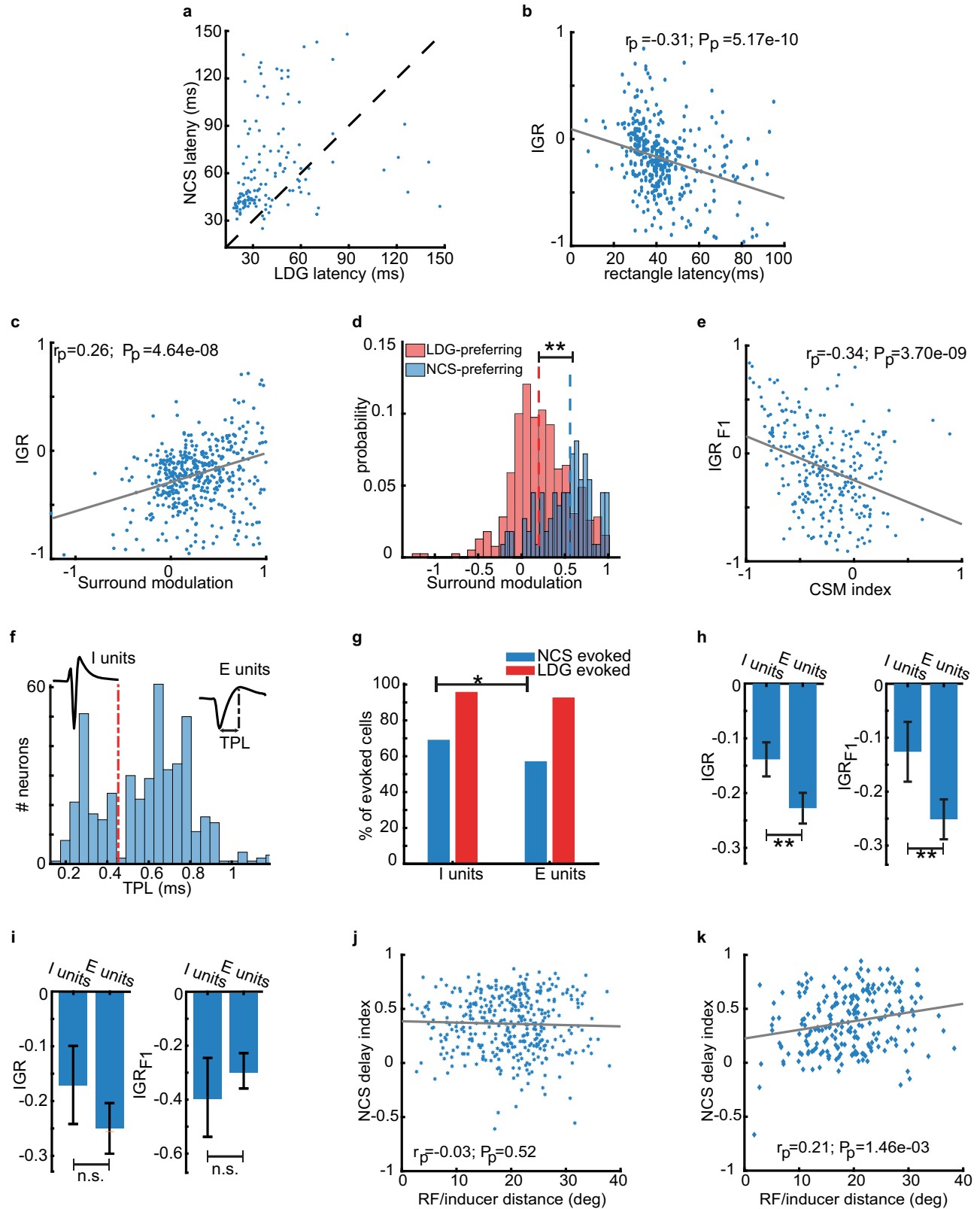

response of two complex cells is shown in Supplementary Fig. 9. Many studies have suggested that complex cells receive bottom-up inputs from simple cells[22,23,31,54,55]. We tested the hypothesis that NCS preferring units would be more likely characterized as complex cells. We separated putative simple and complex cells using a complex-simple modulation (CSM) index, which calculates the power of the F1 component of the grating-entrained response relative to the F0

component (see Methods section, Eq. 3). The CSM index was calculated using responses to LDG stimuli. A CSM index of −1 is indicative of a complex cell, whereas an index of +1 indicates a simple cell. We observed no correlation (Pearson's correlation coefficient, $r = 0.05$, $p = 0.2$) between CSM index and IGR (Supplementary Fig. 10a). We next calculated the IGR index for NCS stimuli relative to LDG stimuli but used the amplitude of the F1 component rather than the average

**Fig. 4 | NCS preferring units respond earlier, receive more surround modulation, and may correspond to complex cells or putative inhibitory neurons.** **a** The response latencies to NCS and LDG stimuli. Points above the dotted line indicate a later response to NCS stimuli relative to LDG stimuli. Each point represents a single unit. **b** IGR plotted against latency of responses evoked by black rectangles. The gray line represents a least squared fit to the data. The *r* value and *p* value (two-sided *t*-test) for Pearson's correlation coefficient are shown. **c** IGR plotted against surround modulation index. Plotting conventions are as in (**b**). **d** Distributions of surround modulation index plotted separately for NCS-preferring units (with higher responses to NSC stimuli) and LDG-preferring units (with higher responses to LDG stimuli). LMEM test: Fstat. = 57.60, *p* = 1.73e-13 (**). **e** The $IGR_{F1}$ is plotted against the complex-simple modulation index. Plotting conventions are as in (**b**). **f** The distribution of extracellular waveform TPL values for all recorded units. The dashed red line on the bimodal distribution shows the intersection of two distinct Gaussian distributions. Example spike waveforms are shown for one example unit from each distribution. Those with short TPL are putative interneurons, and those with long TPL are putative pyramidal neurons. **g** The percent of E units and I units with visually evoked responses to NCS and LDG stimuli. Two-sided Chi-squared test: $\chi^2 = 6.28$, *p* = 0.012 (*). **h** The mean IGR (left) and $IGR_{F1}$ (right) magnitude of I units and E units (error bar = 95% Confidence Interval). LMEM test: *F* = 10.11, *p* = 1.56e-3 (left **); *F* = 10.68, *p* = 1.15e-3 (right **). **i** The mean IGR (left) and $IGR_{F1}$ (right) magnitude of I and E non-overlapping units (error bar = 95% Confidence Interval). LMEM test: *F* = 2.51, *p* = 0.11 (left n.s); *F* = 1.04, *p* = 0.30 (right **). **j, k** NCS delay index plotted against RF-inducer distance for E units (*n* = 385) and I units (*n* = 221), respectively. The gray line in (**h**–**k**) is the least squared fit to the data. The Pearson correlation coefficient and the *p* value (two-sided *t* test) are shown. From (**a**–**h**), *N* = 6 mice. From (**i**–**k**) *N* = 13 mice. Source data are provided as a Source Data file.

stimulus-evoked firing rate. We refer to this version of the IGR index as the $IGR_{F1}$ index (see Methods, Eq. 4). Figure 4e shows that the $IGR_{F1}$ index was negatively correlated with the CSM index (Pearson's correlation coefficient, *r* = −0.34, *p* = 3.7e-9). The same analysis for non-overlapping units led to similar results (Supplementary Fig. 10b, c). This result suggests that units, which are likely to be complex cells, have a larger response entrained to the temporal frequency of the grating for NCS stimuli (an illusory grating) relative to the LDG stimuli (a physically present grating). Thus, the NCS response may involve bottom-up inputs from simple cells in the V1 microcircuit.

Many studies have investigated the functional role of interneurons in the visual system and have shown that interneurons contribute to various neuronal properties, such as orientation and direction selectivities[56–58], and simple/complex RFs[33,59]. We therefore assessed whether V1 interneurons have differential responses to NCS stimuli. We used extracellular waveform characteristics to identify putative V1 interneurons and pyramidal neurons and compare their IGR and $IGR_{F1}$ indices. We determined a putative neuron type using the trough-to-peak latency (TPL) for the average spike waveform of each unit[45,60,61]. The distribution of TPL values was bimodal, thus suggesting two classes of neurons: those with a narrow waveform were considered to be putative inhibitory interneurons (I units, *N* = 146), and those with a wide waveform were putatively pyramidal neurons (E units, *N* = 374). Figure 4f shows the TPL distribution and separation of putative classes of neurons. The average stimulus-evoked responses of each putative cell type are shown in Supplementary Fig. 11. The LDG stimulus-evoked responses in a similar proportion of I units and E units (95% and 92%, respectively; Fig. 4g). Importantly, however, NCS stimuli evoked responses in a significantly larger proportion of I units compared to E units (69% and 57%, respectively; chi-squared test: $\chi^2 = 6.28$, *p* = 0.012). Moreover, the IGR was larger for I units relative to E units (Fig. 4h left; LMEM: *F* = 10.11, *p* = 1.56e-3). We also found that the $IGR_{F1}$ index of I units was higher than that of the E units (Fig. 4h right; LMEM: *F* = 10.68, *p* = 1.15e-3).

However, when we limited this analysis to non-overlapping units, we found that IGR and $IGR_{F1}$ were not significantly different between I units and E units (Fig. 4i). In order to investigate this discrepancy, we estimated the distance between the RF and the center of the closest inducer and calculated its correlation with the NCS delay index (see methods, Eq. 5). The NCS delay index characterizes the NCS response latency relative to the rectangle latency. There was no correlation for E units (Pearson's correlation coefficient: *r* = −0.03, *p* = 0.52, Fig. 4j), but a significant positive correlation was observed for I units (Pearson's correlation coefficient: *r* = 0.21, *p* = 1.46e-3, Fig. 4k). Therefore, as the distance between the unit's RF and the inducer increased, the latency of response to the NCS stimulus increased. These results suggest that putative inhibitory interneurons in V1 could be involved in the spatial spreading and filling-in effect that underlies the appearance of illusory gratings in the NCS illusion.

The V1 microcircuit is characterized by cortical layer-specific cell types and layer-specific intra-V1 synaptic connections (as well as differential extra-V1 afferent and efferent connections). Therefore, we assessed whether NCS responsive units were more prevalent in a specific layer of V1. We used current-source density (CSD) analysis to identify the approximate laminar location of each single unit[45,62,63] (Supplementary Fig. 12a). We compared the IGR index across cortical layers and found that units in layer VI emitted weaker responses to illusory gratings in comparison to all other cortical layers (Supplementary Fig. 12b). We did not observe significant differences in the NCS response latencies in different layers (Supplementary Fig. 12c). Therefore, V1 units that respond to illusory brightness have a bias toward the superficial layers, which receive top-down cortico-cortical feedback, and the granular layer receiving thalamo-cortical input. Overall, our results suggest that brightness illusions activate specific components of the V1 microcircuit: surround suppression, complex cells (though this result was mixed, indicating a nuanced role that does not exclusively favor complex over simple cells), layers I through V, and putative interneurons. Interneurons may play a role in the spatial spread of illusory brightness. These components of the V1 microcircuit and the recurrent serial synapses between interneurons and pyramidal neurons, as well as feed-forward synapses from simple cells to complex cells, may contribute to the delayed response to illusory gratings compared to real ones.

## Top-down feedback from HVAs modulates the V1 single unit response to illusory brightness

Top-down modulation of V1 is thought to play a role in the perception of illusions[36,64]. We hypothesized that feedback from higher levels of visual cortex modulates the NCS-evoked response of V1 units. We tested this hypothesis by optogenetically inhibiting HVAs while recording the V1 single unit responses to the NCS, LDG, and DBC stimuli. Inhibition was achieved by activating parvalbumin-positive (PV+) interneurons in LM and LI, which are analogous to the ventral pathway in primates[65]. In 6 PV-cre mice, we recorded from V1, LM, and LI using a 64-channel silicon probe (30° implantation angle) combined with a tapered fiber for targeted inhibition of LM and LI (Fig. 5a). Figure 5b shows an example multi-unit spiking in response to the rectangle stimuli presented during a RF mapping session. As shown in Fig. 5c, the multi-unit RF of each channel along the probe has a slightly different location than adjacent channels due to the oblique insertion of the electrode. As we move along the probe, the retinotopic RF location on the screen moves in one direction until it enters another visual area where the retinotopic RF location starts to change in the opposite direction (black arrows in Fig. 5c). These reversal points, which indicate the borders of visual areas, were used as an index to specify the area associated with each channel. After mapping the extent of each visual cortex area on the probe, we set the upper bound of optogenetic illumination to limit the light

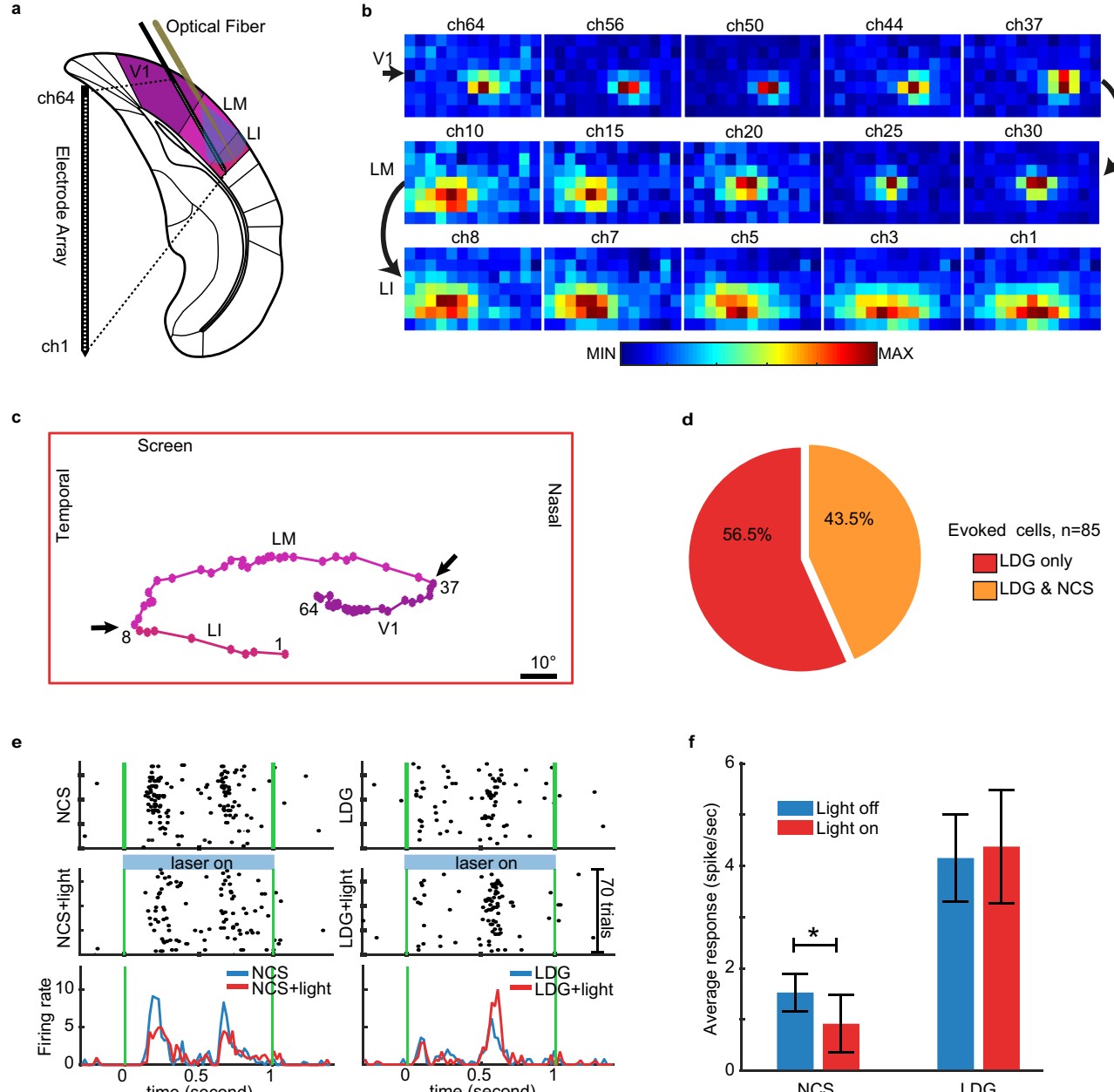

**Fig. 5 | The inhibition of lateral HVAs reduces the V1 single unit response to the NCS stimulus, but not the control stimuli. a** Oblique insertion of a single-shank electrode array combined with optical fiber, targeting mouse lateral visual areas for simultaneous electrophysiological recording and optogenetics manipulation. The silicon electrode array contained 64 recording channels spanning 1275 μm. The optical fiber had a 1.5 mm tapered tip and 200-core diameter. **b** Multi-unit firing intensity maps of a subset of recording channels along the silicon probe showing the RF location on the screen. **c** Tracking the retinotopy of the multi-unit RF centers (colored dots). Black arrows show the reversal points, which determine the borders of visual areas. **d** The pie chart shows the percentage of V1 evoked single units in response to full screen stimuli. **e** Rasters and PSTH of one example V1 unit in response to different stimulus types. The plots show the response to physical grating (LDG stimuli), illusory gratings (NCS stimuli), LDG+light, and NCS+light stimuli, presented at the preferred direction of the unit (225°). The green lines indicate the times of stimulus onset and offset. Note that in some individual units, inhibition of HVAs enhanced the response to LDG stimuli, which is consistent with other studies showing that inhibiting lateral visual areas in mice increases the V1 neuronal response to large stimuli like those used here to study neurons without RF-inducer overlap[85]. **f** The bar plot shows the average firing rate (baseline sub-tracted) across four conditions. (Error bars show the 95% confidence interval of the mean). * indicates $p = 0.04$ and $F = 4.16$ in LMEM test. $N = 6$ mice. **a** was modified from a figure which was published in "The mouse brain in stereotaxic coordinates", Vol 5, Paxinos, George, and Keith B.J. Franklin, Page 97, Copyright Elsevier 2019[86]. Source data are provided as a Source Data file.

leakage onto V1. We ensured that V1 activity was not directly inhibited by the light (see Methods). As a result of optogenetic manipulation, we observed that laser emission significantly affected the NCS response of 78.94% of units in LM and 85.48% of units in LI (either increased or decreased activity). Supplementary Fig. 13 presents the NCS and LDG responses of individual LM and LI units in the presence and absence of optogenetic stimulation.

We constrained our analysis to 85 V1 units (out of 653) that did not have RF-inducer overlap. 56.5% of these units responded only to the LDG stimuli and 43.5% responded to both the LDG and NCS stimuli (Fig. 5d). After inhibition of LM and LI, the V1 unit response to the NCS stimulus was reduced (LMEM: $F = 4.16$, $p = 0.04$). Example units are shown in Fig. 5e. At the population level, inhibition of HVAs diminished the V1 unit response to illusory brightness without modulating the response to real

gratings (Fig. 5f). The responses of individual single units are shown in Supplementary Fig. 14. These results indicate that top-down input from HVAs to V1 exerts a modulatory influence on V1 specifically in the context of processing stimuli that evoke illusory brightness.

## The pupil responds in opposing directions for illusory decrements in brightness compared to real increases in brightness

A most intriguing question in studying the neural correlates of visual illusions in animals is whether they are related to the "perception" of the illusion. In the absence of a system (i.e., language) for reporting subjective, first-hand perceptual experience, the perception of illusory brightness can be probed in pre-linguistic subjects (e.g., human babies) and subjects lacking language (e.g., animals) using pupillometry[66]. In humans, subjective perception of brightness or darkness evokes a pupillary response[37–40]. For instance, images we subjectively interpret as bright objects, such as a picture of the sun, will evoke pupil constriction relative to control images that are not interpreted as bright even though both images have the same physical luminance[37]. Brightness illusions that cause pupil constriction in humans also do so in rats[41]. Therefore, the pupillary response can be used as putative and indirect evidence of perceptual report in pre-linguistic human subjects and animals, alike.

We hypothesized that the pupil would dilate after the NCS stimulus because it is perceived by humans to have "darker than black" gratings. On the other hand, we predicted the opposite pupil response (constriction) after the LDG stimulus, which is perceived as an increase in brightness by human subjects. As a control, we expected the DBC stimulus to not evoke a pupillary response because it was identical to the NCS stimulus, but contained occluders that block the perception of illusory darkness in humans. We measured pupil size in 6 mice. In support of our hypothesis, the pupil dilated in response to the NCS stimulus, constricted in response to the LDG stimulus, and remained stable for the DBC stimulus (Fig. 6a). We calculated a pupil dilation index by subtracting the baseline pupil size (averaged over the 0.2 s window before stimulus onset) from the unnormalized pupil size (averaged over the 1 s window of stimulus presentation) and dividing it by the sum of the two. A positive value is a dilation, a negative value is a constriction, and a value of approximately zero is no change from baseline. The magnitude of the dilatory response to the NCS stimulus

was as strong as the constriction to the LDG stimulus, as assessed using the pupil dilation index (Fig. 6b). Note that the luminance for NCS and DBC reduces by 0.82% during stimulus presentation compared to pre-stimulus luminance. Critically, despite this similarity in luminance reduction, the pupil responses to these two stimuli were markedly different (Fig. 6). This disparity in pupillary response, despite similar luminance reductions, suggests that the pupil reactions are not solely due to luminance changes. Conversely, luminance for the LDG stimulus increases by 23.35%. Although the luminance change in LDG, versus NCS, is 28 times higher, the max pupil response in these two conditions is similar (-8% constriction in response to LDG and ~6% dilation in response to NCS). These results demonstrate that luminance differences do not explain the pupil response magnitude. The opposing responses of the mouse pupil to the NCS and LDG stimuli provide a reflexive behavioral report that indirectly supports the perception of the illusory grating by mice, as it does in humans.

## Discussion

Illusions are a powerful tool for studying the neural correlates of subjective perception. Studies of the V1 single neuron response to illusory brightness are limited[7,8], have produced inconsistent results with respect to fMRI studies[9–14], and have not investigated properties of the cellular microcircuitry or their dependence on top-down modulation from higher visual cortex area. Major theoretical views depend on properties of microcircuits and feedback[35,36,47,48] and therefore, experimental data at the microcircuit level and manipulation of feedback from HVAs are critical for supporting theory at the single neuron level. In addition to the limited single neuron studies in animals, fMRI studies of human V1 BOLD signal responses to different features of illusory surfaces, specifically illusory fill-in in NCS[9] and the Cornsweet illusion[10], have been unable to discriminate between the perceptual experience of brightness, color, or other aspects of stimulus[12,14,64]. Moreover, only limited extrapolation can be made about neuronal activity from the fMRI BOLD signal. Overall, it is controversial whether V1 contributes to the perception of illusory brightness[7,16–18], and thus, the neural correlates of illusory brightness remain largely unknown.

Here, we took advantage of NCS – previously only demonstrated in humans[9,21] – to produce a brightness illusion that forms an oriented grating. We used an achromatic version of NCS stimuli to probe the

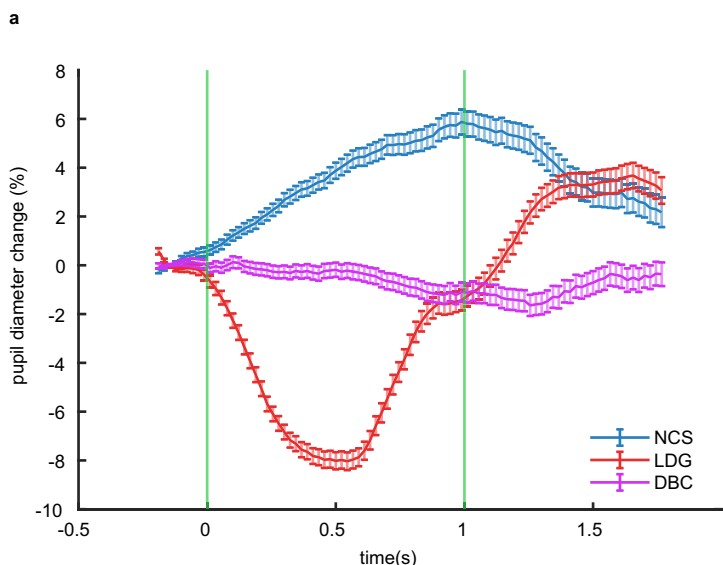

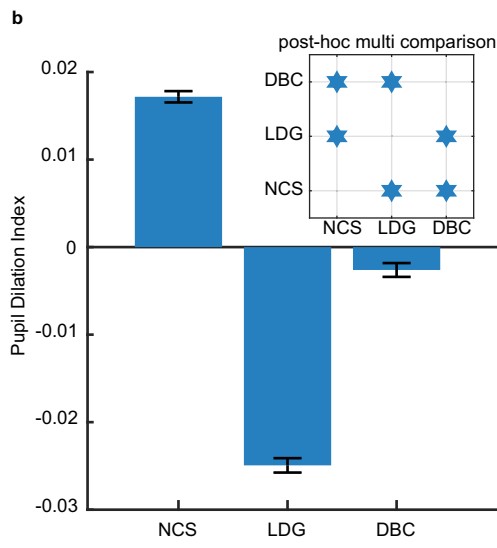

**Fig. 6 | The mouse pupil responds to illusory brightness. a** The average pupil size over time normalized to baseline. The green lines indicate stimulus onset and offset. The error bars show standard error of the mean (SEM). **b** The mean and SEM of the pupil dilation index for each stimulus type. The inset shows the results of Kruskal-Wallis test H(2) = 2488.83, $p < 0.0001$ and the post-hoc multi-comparison (Tukey's Honestly Significant Difference Procedure) of the dilation index between each of the stimulus types. Stars represent the significant difference ($p < 0.0001$). $N = 6$ mice. Source data are provided as a Source Data file.

neural correlates of illusory brightness in mice. The specific use of gratings allowed us to compare multiple properties of the neuronal response between illusory and physical gratings, including drift direction (angle) tuning properties, entrainment to the temporal properties of the grating (i.e., the F1 component of the neuronal response), and phase shifting to illusory and physical gratings with opposing phase. Using this design, we show that mouse V1 single units respond to the illusory drifting grating evoked by NCS stimuli even if there is no direct RF stimulation. Critically, V1 single units did not respond to control stimuli in which pixel-wise changes in physical luminance are matched to the NCS condition but illusory brightness is blocked. These control stimuli do not evoke the perception of illusory brightness or illusory gratings by human observers (see Supplementary Movie 2). We found that the neuronal tuning properties are similar for real gratings (LDG stimuli) and illusory gratings, which suggests that NCS stimuli evoke neuronal responses characteristic of those to actual gratings. Importantly, by presenting illusory gratings and physical gratings with a 180° spatial phase shift, we show that V1 neurons respond to the spatial phase properties of the illusory gratings and, therefore, track the illusory brightness perceived by human subjects. Collectively, these results are strong evidence for the response of V1 single units to illusory brightness.

Given the lack of direct demonstration of a V1 single unit response to illusory brightness, it is unsurprising that there has been no study characterizing the V1 microcircuits involved in processing of brightness illusions. Our paradigm, translated from humans to mice, demonstrates a V1 single unit response to brightness illusions and allowed us to uncover a few aspects of the V1 microcircuit that are responsive to illusory brightness. These include surround suppression, complex cells, and putative fast-spiking interneurons. We found that V1 neurons with greater surround suppression effects have a greater response to NCS stimuli. At a mechanistic level, this finding is consistent with multiple potential interpretations. For instance, it may support the notion that top-down feedback onto V1 plays a role in illusory brightness perception because optogenetic studies have shown that surround suppression in V1 depends on feedback connections[25,67]. Therefore, the activation of surround suppression by the NCS stimuli could indicate the engagement of feedback. However, optogenetic studies have also shown that surround suppression depends on intra-V1 horizontal connections[52] and therefore, our result may also be interpreted as a sign of activation of specific aspects of the intra-V1 microcircuit. On the other hand, several studies have demonstrated surround suppression in the lateral geniculate nucleus (LGN), suggesting surround suppression could be partially inherited from LGN through feed-forward connections[68–71]. Although the intra-V1 and extra-V1 neuronal connections involved in illusory brightness processing remain unclear, our work demonstrates that surround suppressed cells are involved in processing of NCS stimuli.

We also evaluated the propensity for complex cells to respond to illusory gratings as a means of characterizing the involvement of various aspects of the V1 microcircuit in processing of the NCS stimulus. Complex cells receive input from simple cells, which may be thought of as a "lower" hierarchical level in the V1 microcircuit[23,29–32,54]. Intriguingly, we found no correlation between IGR and the CSM index, showing that complex and simple cells seem to have a similar propensity for illusory and real gratings. On the other hand, we observed that complex cells had a higher F1 response to the illusory grating. This discrepancy (IGR versus IGR calculated on the F1 component) may be related to the fact that the F1 component encodes the perceptual content of the drifting grating. However, it is clear that the role of complex and simple cells in illusion processing currently needs to be clarified and requires further study.

It has been shown that interneurons play a crucial role in visual perception[56–58,72–75]. We assessed the responses of putative interneurons and pyramidal neurons and found that putative interneurons were more responsive to NCS stimuli than LDG stimuli. Therefore, V1 interneurons may contribute to the processing of illusory brightness. Interestingly, we found that their latency to respond to the NCS stimulus increases with distance between the RF and the inducer, which could indicate a role in the filling-in effect. The NCS stimulus depends on an illusory filling-in of space between the inducers. As the distance from the inducer increases, the neuron's RF overlaps more with the area "filled in" by the illusory brightness change. The increased response latency at filled-in locations may be due to progressive integration or interpolation across the visual field. We found that it is primarily interneurons that respond in this manner and may, therefore, be involved in filling in (at least in this particular illusion). However, it is currently unclear how interneurons and synaptic inhibition might generate filling-in. It is possible that network-level effects, such as interactions with excitatory neurons or disinhibition (via interneurons inhibiting other interneurons), play a role in filling in when the brain confronts ambiguous or incomplete visual cues.

The predominant, yet unproven, theoretical view for the neuronal mechanism of subjective perception is that it is critically dependent upon late-stage synaptic feedback to V1 neurons that occurs after an initial feed-forward pass through V1[36,64,76]. One prediction of this theory is that the V1 neuronal response latency should be delayed for illusory stimuli relative to real stimuli because of the time required for activation of additional synapses extrinsic to V1. Indeed, in line with the prior work on response latencies in humans[47], macaques[35], and mice[36], we found that the neuronal response to the illusory grating was delayed relative to a real grating. Our findings support the theoretical notion that the V1 neuronal response to illusory gratings cannot be only feed-forward and driven by physical changes in the RF. Here, we performed a critical test of this theory by optogenetically inhibiting the HVAs during the presentation of an illusory brightness stimulus and showed that this diminished the V1 single unit response to the illusory grating. Our findings provide a causal test of the role of feedback from HVAs onto the V1 processing of brightness illusions. While our results suggest a key role for top-down modulation from HVAs, it is also important to note three caveats. First, we could not record from or precisely manipulate neurons in HVAs with RFs overlapping with NCS-responsive V1 neurons due to using a single multi-electrode shank covering multiple cortical areas. Therefore, we cannot conclude that HVAs provide targeted top-down modulation to retinotopically-matched V1 neurons, only that HVAs provide a general modulation of the V1 response to illusory stimuli. Second, this evidence supporting top-down modulation does not exclude a role for bottom-up inputs, such as responses to illusory gratings already at the thalamic inputs to V1. Critically, the mouse visual paradigm presented here can be used for thalamic recordings to test the role of bottom-up inputs. Lastly, while the optogenetic modulation provides strong evidence for top-down modulation, it is essential to note that indirect electrophysiological measures of top-down modulation (i.e., the layer-specific response) and serial synaptic activation (i.e., the rectangle latency) do not clearly indicate that the NCS response is due to top-down modulation or occurring at a later stage of serial synaptic activation. Collectively, the results suggest a complex interplay between bottom-up and top-down processing in visual perception beyond the traditional conception of visual cortex processing as a strictly hierarchical system.

Our results demonstrate, at a different level of detail compared to fMRI BOLD, the single cell and V1 cellular microcircuit-level neuronal correlates of visual processing of illusory brightness. Moreover, we show its dependence on feedback from HVAs. These findings were made possible by translating an illusory brightness stimulus from studies in humans into a compatible format for electrophysiology in mice. However, one limitation of this experimental paradigm is that it is challenging to make strong claims about how single unit activity relates to perception. Therefore, we measured the reflexive pupil response to the illusory and real gratings as an indirect measure of

perceived luminance, as has been done previously in human subjects[37] and in rats[41]. Our findings that the mouse pupil responds in opposing directions for these two stimuli, which humans perceive as getting darker or brighter than the background, are potentially consistent with mice and humans similarly perceiving these stimuli. While future work is needed to support this limited claim (e.g., with an overt behavioral report in a brightness discrimination task), our findings do unravel, at the single-cell level, the V1 processing (or "sensing") of illusory brightness which may be seen as distinct from attaching an interpretation or meaning to stimuli as being brighter ("perception")[77]. By building on our indirect measure of perceived luminance, future work can use overt behavioral reports to establish a link between V1 single cell activity and subjective perception of brightness illusions.

## Methods

Experiments were performed on awake head-fixed adult mice on a disc. The local authorities (Regierungspräsidium Tübingen) approved all animal procedures and the procedures were done in compliance with EU Directive 2010/63/EU (European Community Guidelines for the Care and Use of Laboratory Animals). Data acquisition was done through several electrode penetrations in both hemispheres of 19 C57BL/6 or PV-Cre mice (homozygous for the PV-Cre genes, B6;129P2-Pvalbtm1(cre)Arbr/J). The sex specification of the mice and the number of performed experiments on each mouse are presented in Supplementary Tables 1–3. Mice were housed with sibling cagemates on a 12-h light/dark reverse cycle. Humidity was between 40% and 60%, and the temperature was $22 \pm 1\,°C$. After head-post surgery, mice were singly housed.

### Surgical preparation

Mice were induced by 2.5% of isoflurane during surgery and maintained at 1–2%. Also, Atropine (Atropinsulfat B. Braun, 0.3 mg/kg) and Buprenorphine (0.1 mg/kg) were administered via subcutaneous injections to reduce bronchial secretions and as analgesics, respectively. The scalp was sterilized and opened to expose the lambda and bregma sutures. A lightweight head-post was installed onto the skull using an adhesive primer and dental cement (OptiBond FL primer and adhesive, Kerr dental; Tetric EvoFlow dental cement, Ivoclar Vivadent). A small well was built around the exposed area using dental cement. Two silver wires were implanted between the dura and skull over the frontal lobe as ground references for extracellular recordings. Then, the skull was covered with Kwik-Cast (WPI). The post-surgery analgesic (Flunixin, 4 mg/kg) continued to be administered every 12 h for three days, and antibiotics (Baytril, 5 mg/kg) were administered for five days. After recovery, animals were habituated to head-fixation and placed on a disc for three days (0.5 h/day). On the fourth day, a small craniotomy (1 mm²) was drilled above the V1 at 2.5 mm laterally and 1.1 mm anterior of the transverse sinus[78] under general anesthesia. Electrophysiological recordings were started one day after craniotomy surgery and continued on consecutive days for as long as the neuron isolation remained of high quality. The craniotomy was covered with Kwik-Cast after each recording.

### Electrophysiological recordings

Mice were head-fixed on a disc and allowed to sit or run on it in a dark and electromagnetic isolated room. In a group of six mice, a 32-channel linear silicon probe (Neuronexus, A1x32-5mm-25-177-A32) was penetrated the V1 perpendicularly to a depth of ~900 μm below the brain surface (the depth of electrode insertions is presented in Supplementary Table 1). In 13 mice, a 64-channel silicon probe (CambridgeNeuroTech, H9 or H3) was inserted with an oblique angle (~30°) penetration distance ~1400 μm below the brain surface. Electrical signals were amplified and digitized at 30 kHz by the Cerebus data acquisition system (v7.0.4; Blackrock Microsystems LLC) or RHD recording system (Intan Technologies) and Open Ephys GUI (v0.4.4 or

v0.4.6). NPMK (VS.0.0.0; BlackrockNeurotech) was used to read recorded data. A photodiode was attached to the lower right corner of the screen to capture the exact stimulus onset from a white square synchronized to the stimulus presented. A rotary encoder (US Digital, MA3-A10-125-B) connected to the disk converted the disk angle to a voltage between 0 V and +5 V, and the analog signal corresponding to the axis rotation was recorded as input into the acquisition system at 30 kHz.

### Visual stimulation and experiment design

Stimuli were projected onto a gamma-corrected LED monitor (Dell U2412M, 24 inches, 60 Hz) placed 15 cm in front of the animal's eye. Visual stimuli were programmed and generated in MATLAB (Math-Works, Inc.) and Psychophysics Toolbox Version 3 (PTB-3). To obtain the RF map of recorded neurons, black rectangles (~15° widths) were presented on a gray background with a duration of 100 ms and an interval of 100 ms in different locations of 8 by 13 grids. The duration of the RF mapping session was 20 min. After this section, the response to each rectangle was extracted by an analysis of multi-unit activity (MUA). Then, the center of the MUA RF was estimated by the best fit of a two-dimensional Gaussian to the MUA activities. Subsequent target stimuli were presented on a gray background at the estimated RF center. To obtain the size-tuning curve, circular patches of drifting grating (spatial frequency 0.05 cycles/degree, temporal frequency 3 Hz) with different sizes (2.5, 5, 10, 15, ..., 45°) and two drifting directions (rightward and upward) were presented. The duration of the stimulus presentation was 666.7 ms with a 500 ms interval, and the duration of the whole session was 25 min.

We presented three types of drifting grating stimuli for the neon color session. (1) Neon color spreading (NCS) consists of nine patches of white concentric circles (0.1° thickness) as inducers (each patch had three circles), arranged on a three-by-three virtual grid on a black square (35° widths). The diameter of the inducers was 3, 6, and 9°, respectively. At each frame, the intersection of concentric circles and a drifting grating (spatial frequency = 0.05 cycle/degree and temporal frequency = 2 Hz) was replaced with gray segments, resulting in the "darker than black" illusory grating (Fig. 1a, Supplementary Movie 1). (2) Diffusion-blocked stimulus is a control condition with exactly the same pixel-wise changes as NCS, while each inducer circle is sandwiched by two white circles (0.4° thickness). The added circles constrain the gray filling-in and reduce the illusory effect (Fig. 1c, Supplementary Movie 2). (3) Luminance-defined grating (LDG) is defined as gray grating moving on top of inducers (Fig. 1d, Supplementary Movie 3). These three types of drifting grating were generated in eight directions, making 24 conditions. These stimuli were presented with a duration of 1 s for 70 trials in a pseudo-randomized order. We also generated a fullscreen version of the same sets of stimuli and used them in our experiments with the following specifications. The thickness of white concentric circles (i.e., inducers) was 0.4°, and their diameter was 8, 16, and 24° in a batch of 3. The arrangement of concentric inducers was on a honeycomb structure, providing a maximal illusory area with a diameter of 39° between each patch of inducers. The thickness of blocking circles for fullscreen DBC was 0.8°.

### Viral injection and optogenetics manipulation

For viral injection, we drilled a small craniotomy window (<1 mm) above the LM region (from bregma, AP: −4 mm, LM: 4.1 mm). We used a glass micropipette containing undiluted Cre-inducible DIO-AAV4 (AAV-EF1a-DIO-hChR2(H134R)-EYFP-WPRE-pA) to perform the injections at a rate of 6 nl/s using Nanoject III (Drummond Scientific). The injections were carried out at four depths, 0.2, 0.4, 0.6, and 0.8 mm below the cortical surface, with 40 nl of the virus injected at each depth. We started the injections at the deepest spot, and after each

injection, the glass pipette was left in place for 5 min before being pulled up. After the injection into both hemispheres, we covered the skull with Kwik-Cast (WPI). The animals were observed to regain consciousness within approximately 30 min following the surgical procedure. After the surgery, they were kept under monitoring for three days. We started the optogenetics experiments four weeks after viral injections.

In order to suppress LM and LI, we expressed the Channelrhodopsin-2 (CHR2) gene in parvalbumin-positive (PV+) interneurons. We stimulated the specific region using blue light (473 nm laser from Laserglow Technologies). To deliver spatially selective light, the laser was connected to an optomechanical tool equipped with tapered fibers (ThetaStation OptogeniX). This tool allowed for precise control of light emission through the manual operation of a micrometric screw, which determined the specific sub-region of a Lambda fiber that emitted the light.

For simultaneous optogenetics manipulation and electrophysiology, we used a silicon probe combined with tapered optical fiber (H3 probe, Cambridge NeuroTech; lambda-b fiber 200 core, 0.66 NA, emitting length 1.5 mm, Optogenix). We calibrated the power level for each depth at 100 μW by adjusting the power at the end of the patch cord (before implantation). The optogenetics session included electrophysiology recording and the presentation of NCS and LDG with and without optogenetic manipulation.

The depth of light emission was adjusted after the receptive mapping session and identifying the borders of visual areas by finding reversal points of multi-unit RF centers on the screen[65] (Fig. 6c). This adjustment aimed to avoid direct suppression of spontaneous activities in the primary visual cortex (V1). During the light stimulation, a constant one-second pulse was delivered synchronously with the presentation of the visual stimulus. The optical setup was controlled by Psychophysics Toolbox Version 3 (PTB-3) and custom code in Matlab, which communicated with a modified sound card using a TTL signal.

## Data analysis

For spike detection and clustering, we first concatenated the recorded data in all three experiment sessions (i.e., RF mapping, size tuning, and neon color). We then used the Kilosort algorithm, a template matching algorithm written in MATLAB for spike sorting, with the default parameters[79]. A manual clustering followed this for further merging, splitting, and choosing isolated clusters using template-gui (phy v2)[80]. All further analyses were done in MATLAB (v R2018a-R2020a) using built-in functions. The peristimulus time histogram (PSTH) was initially calculated with a resolution of 1 ms and smoothed by a moving average window of 2 ms. We estimated single-unit RFs by fitting a two-dimensional Gaussian function to their spiking activities in the RF mapping session. To extract visually evoked neurons and estimate their onset latency, we assumed that the spontaneous spiking activity prior to stimulus presentation follows a Poisson distribution[49]. By fitting a Poisson distribution to 300 ms prior to stimulus onset, the spontaneous firing rate λ was estimated. If the spiking activity after stimulus onset deviates from the background Poisson distribution to a particular level in three consecutive bins (a probability of $p < 0.01$ for the first two bins and $p < 0.05$ for the third bin), the neuron was considered as an evoked neuron, and the corresponding time for the first bin is considered as a response latency of the neuron[49]. The preferred angle of cells is defined as the stimulus direction with the maximum response. To calculate changes in the preferred angle of neurons in NCS and LDG, first, we captured the direction tuning curve of the neuron by taking the average response of cells during the stimulus presentation (one second). Then, we interpolated the tuning curve with the spline method to get a more precise estimation of the preferred angle.

We have quantified the illusory grating response for each neuron as follows.

$$IGR = \frac{R(NCS) - R(LDG)}{R(NCS) + R(LDG)} \tag{1}$$

Where R is the average response of the neuron to the stimulus over the presentation period. A positive IGR represents a higher response to NCS and vice versa.

We defined the surround modulation index as

$$surround\ modulation = \frac{\max(R(<30°)) - R(45°)}{\max(R(<30°))} \tag{2}$$

The surround modulation index is negative for facilitative cells and positive for suppressive cells.

We implemented the Fast Fourier Transform (FFT) on the PSTH to extract a temporal component of neuronal responses. The F1 component of the response is the power for 2 Hz, which is the same as the temporal frequency of the stimulus, and the F0 component is the average response.

The complex-simple modulation index was calculated using the following equation for every single unit:

$$CSM = \frac{F1 - F0}{F1 + F0} \tag{3}$$

Where F1 is the power of 2 Hz frequency, and F0 is the average firing rate in response to LDG stimuli. Neurons with a positive CSM index have a phase-locked response to the temporal frequency of drifting grating and are classified as simple cells. The phase-locked response is due to the separated excitatory and inhibitory subregions in their RF. A negative CSM index indicates the degree of spatial invariance in the RF and a lack of phase-locked responses.

We calculated the relative amplitude of the F1 component (IGR$_{F1}$) as in the following,

$$IGR_{F1} = \frac{F1(NCS) - F1(LDG)}{F1(NCS) + F1(LDG)} \tag{4}$$

where F1(NCS) and F1(LDG) are the amplitude of the F1 component of the response to NCS and LDG respectively.

We assessed the delay of NCS responses compared to the response to black rectangles for every single unit using the following equation.

$$NCS\ dely\ index = \frac{Latency(NCS) - Latency(rectangle)}{Latency(NCS) + Latency(rectangle)} \tag{5}$$

Where the Latency(NCS) is the response latency to NCS stimuli and Latency (rectangle) is the latency of response to the contrast change of black rectangles within the RF, presented during the RF mapping session.

To classify cells into two groups of putative inhibitory and excitatory cells, we fitted two Gaussian functions to the histogram of TPL, which shows a bimodal distribution. The intersection point of two Gaussian curves was selected as a threshold to classify putative I/E cells (Supplementary Fig. 15).

To extract the local field potential (LFP) signal, the extracellular signal was filtered within the 0.1 to 150 Hz frequency range using a Butterworth bandpass filter (Matlab "butter" and "filtfilt" functions) and downsampled at 1 kHz. To compute the CSD, we took the discrete second derivative of the LFP signal across the electrode sites. The resulting CSD map was then interpolated to obtain a smooth and continuous signal representation.

To calculate disk speed, we downsampled the rotary encoder signal to 300 Hz and then converted the voltage data to disk angles. Voltage from 0 V to 5 V was mapped to 0-to-360°. The linear speed of the mouse was estimated by calculating the discrete differential of the disk angle over time and multiplying it by the disk diameter. We classified trials into two groups "run" and "still" trials. A trial was considered as run if the average mouse speed was higher than 1 (cm/s) in the first 500 ms of the trial (from 300 ms before stimulus onset).

To calculate the overlap ratio, we first extracted the single unit responses to the presented rectangles during the RF mapping session. This resulted in a two-dimensional heatmap of the firing rate for each single unit. We scaled up the resolution of the heatmap using the Nearest-neighbor interpolation method. We masked all the pixels with an activity lower than three standard deviations to infer the RF of units. Supplementary Fig. 16 illustrate the masked activity of 10 non-overlapping RFs. Then, we fitted a 2-dimensional Gaussian function on the masked firing rate of the units. Finally, we focused on quantifying the overlap between the RFs and the inducers. We fitted an ellipse at the Full Width at Half Maximum (FWHM) of the 2-dimensional Gaussian curve to achieve this. We then calculated the overlapping area between this ellipse and the largest surrounding inducers. This precise calculation allowed us to quantify the degree of overlap and assess the extent to which the inducers influenced each unit's RF.

We used the Linear Mixed Effects (LME) model as the primary statistical analysis unless otherwise specified. We incorporated two levels of hierarchy in the LME model to account for the nested structure of our data: the mouseID, representing individual mice, and the recording days, representing different days of data collection. For comparison of more than two groups of data, we used the nonparametric Kruskal-Wallis test which was followed by a post-hoc multiple comparison with Tukey's honestly significant difference correction. For directional statistics, Rayleigh's test has been implemented. To calculate Pearson's correlation and its significant level ($p$ value), we used the Matlab "corr" function, which etimates the $p$ values using a two-sided $t$ test. We used the CircStat Toolbox[81] for statistical analysis of circular data.

**Pupillometry data acquisition and analysis**
Videos were captured from the mouse eye at a rate of 45 frames per second using a near-infrared camera (Allied Vision, G-046B) with a variable zoom lens, fixed 3.3x zoom lens, and 0.25x zoom lens attachment (Polytec, 1-60135, 1-415 62831, 6044). The video was acquired via a GigE connection using the MATLAB image processing toolbox. With each video frame, the camera provided a TTL pulse recorded directly into the neurophysiology recording system to align pupillometry data with stimulus presentation in further analysis. We used near-infrared light (Thor Labs LED, M850L3 and Thor Labs Collimation optics, COP4-B) placed near the mouse eye (-30 cm) for illumination, and whole experiments were done in a dark chamber with no external light.

We employed DeepLabCut, a computer vision software, to analyze the pupillometry videos and extract pupillometry data. The training dataset was created by manually labeling and outlining the pupil on 150 randomly selected frames from recorded videos of various mice. Data augmentation techniques, specifically the "imgaug" option, were utilized to enhance the training dataset. We utilized the "efficient net-b0" network and trained it for 500,000 iterations to analyze the recorded videos and extract pupil size measurements. To enhance the quality of the output results, we further applied the outlier extraction technique (also implemented in DeepLabcut) to refine the pupillometry data[82]. We then chunked and aligned the extracted pupil size with the stimulus onsets using the TTL signal of the camera. The pupil diameter was subtracted and divided by the average pupil size over 300 ms prior to the stimulus onset.

**Panel figures preparation**
Data was visualized using Matlab, and Adobe Illustrator 2020 was used to arrange panel figures.

**Reporting summary**
Further information on research design is available in the Nature Portfolio Reporting Summary linked to this article.

## Data availability
Data required to generate figures are provided with this paper and are available at: https://doi.org/10.6084/m9.figshare.24635439.v5[83]. Note that these are the required data to reproduce the figures in this publication using codes provided in the 'Code availability' section below. I addition, continuous raw data (Electrophysiology signal recorded at 30 kHz) is also available upon request from the corresponding authors. Source data are provided with this paper.

## Code availability
The codes for analyzing and reproducing the main figures are available online at GitHub: https://doi.org/10.5281/zenodo.10635663[84].

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

## Acknowledgements

This work was funded by the Max Planck Society and the Helsinki Institute of Life Science at the University of Helsinki (NT).

## Author contributions

Conceptualization – M.W.; Data acquisition and curation – A.S., K.W., G.N.; Formal analysis – A.S., M.W., N.T., A.B.; Methodology – A.S., M.W., N.T.; Project administration – M.W.; Supervision – M.W., N.T., A.B.; Visualization – A.S., M.W., N.T.; Writing – A.S., M.W., N.T., A.B.; Resources – N.K.L.

## Competing interests

The authors declare no competing interests.
