## [Peer Review File · Nature Communications]

Brightness illusions drive a neuronal response in the primary visual cortex under top-down modulationREVIEWER COMMENTS

Reviewer #1 (Remarks to the Author):

The manuscript by Saeedi and colleagues describes the neuronal correlates of illusory brightness in V1. The aim of the manuscript is valuable, as it complements the current literature on perceptual illusions in rodents, a model that can help establish the neuron-level mechanisms of illusory percepts.

The presented analyses are generally well done, but I have some concerns, mainly about the statistical approach and the conclusions of the authors.

MAJOR

1) The authors claim that the fact that information about illusory brightness is encoded in V1 is evidence that the illusion works as well in mice as in humans. However, in the absence of behavioral evidence, this claim remains in my opinion a bit speculative. I would recommend that the authors carefully check the wording of their main claim, and add a discussion point about the lack of behavioral confirmation of the perceptual illusion.

2) The data originates from 32 recordings in 6 mice (if I interpreted the methods correctly). However, the statistical analyses do not take this into account, but consider each neuron as an independent unit, from the statistical point of view. The authors should rather perform multi-level statistics, taking into account the nested nature of their data.

MINOR

1) In view of the large number of recorded units, it would be interesting if the authors could estimate whether effects vary across cortical layers. This would be especially relevant to validate the claim that encoding of illusory brightness originates from top-down modulation.

2) Mice were sitting on a disc, which I assume could be moved by animals when they attempted locomotion. If this is the case, did the authors assess when animals were moving (i.e. in a state of high arousal) and did they exclude such epochs? It would be important at least to better specify the experimental setup and, if possible, separately analyze periods of motion, in view of the large effect of arousal on sensory processing.

Reviewer #2 (Remarks to the Author):

The manuscript by Saeedi and colleagues titled 'Mouse primary visual cortex neurons respond to the illusory "darker than black" in neon color spreading' examines whether the primary visual cortex (V1) in mouse contains neurons that respond to the illusion of drifting gratings. Specifically, the illusion used here is referred to as 'neon' in the human literature. The development of the 'neon' stimulus is novel and exciting. The authors used this stimulus to convincingly demonstrate that mouse neurons in V1 respond to the illusion as if there was a grating present. The characterization of responses to classic gratings and the illusion for the same neurons is informative, and the discovery that the temporal dynamics of the illusion response is delayed is of interest because it implies that circuit elements are 'filling-in' a feature not present in the actual stimulus. To be of broad interest, it would be useful if the authors could provide further evidence testing their hypothesis that top-down inputs contribute to the illusion responses, or alternatively demonstrate that top-down inputs do not contribute- either outcome would be of interest, and to pin-point the layers in which the illusion response is observed most prominently. Overall, although the new stimulus is novel and has potential to lead to interesting discoveries, it is unclear how this study advances our understanding of visual processing.

Major

1. Significance of the manuscript would be improved if the authors identified the level in the visual hierarchy in which responses to the illusion first appeared. For example, are the layer 2/3 responses inherited from layer 4 or is layer 2/3 the first site of emergence?
2. Do the authors have evidence disentangling the two possible interpretations, intra-V1 versus extra-V1? For example, if V2 is inactivated do the responses to the illusion weaken?
3. The extent to which complex cells in the mouse preferentially receive top-down feedback is unknown, therefore the authors have not met the bar to claim that their data indicate top-down input plays a role in the observed response to the illusion stimulus. What is the evidence in mouse that complex neurons preferentially receive top-down inputs over for example, simple cells?

4. The discussion on the role of interneurons is opened-ended, and vague. Does the authors' argument require that interneurons are preferentially contacted by top-down feedback over specific classes of excitatory neurons? If so, please provide evidence, either experimentally or in literature.

5. The rationale for considering preference for NCS versus LDG in Figure 3c is unclear.

6. It is unclear how narrow-spiking neurons relate to surround modulation, given it has been shown that somatostatin neurons, which generally have a regular-shaped waveform, contribute to surround modulation.

7. If the recordings were done in awake animals, please make that clear. Given the hypothesis is that top-down feedback contributes to the illusion responses, it would be problematic if the recording were done in an anesthetized prep.

Minor

1. Title could be improved to capture a broader audience. As written, I think only visual neuroscientists will relate. The use of 'color' in the title and discussion is a little confusing. Although inspired by color spreading studies in humans, there is no color used here.

2. Depth of recoding from pia surface should be reported.

3. More details on the receptive field mapping should be incorporated into the analysis of the illusion responses, to ensure that the main effect holds if only neurons that were stimulated in the center of their receptive field are included in analysis.

4. Online site for code availability is missing.

5. More examples of stimulus responses would be useful.

6. The number of animals used in each plot is unclear.

7. The density of simple cells differs across layers, please account for this in the analysis and interpretation.

Reviewer #3 (Remarks to the Author):

The manuscript by Saeedi et al. demonstrates that neurons in mouse visual cortex respond to a visual stimulus (neon color spreading stimulus, NCS) that induces perception of an illusory drifting grating in humans. The study reports that NCS-responding neurons have similar orientation and direction tuning to NCS and real luminance-defined gratings (LDG) stimuli, and have F1 responses that correspond to the spatial phase features of each stimulus. Neural preferences for NCS vs LDG correlate with other properties such as surround suppression, relative F1 vs F0 response (simple/complex), and spike waveforms. NCS-preferring neurons also exhibit greater surround suppression and have more complex-cell-like properties, and cells with fast-spiking waveforms had greater preference for NCS. From these neural responses, the study puts forth two main conclusions: 1) mouse visual cortical neurons are responding to NCS stimuli and thus respond to illusory luminance changes (as perceived by humans) and 2) NCS-preferring neurons are positioned higher in the visual hierarchy, allowing them to receive more top-down inputs which could contribute to their enhanced responses to NCS stimuli. The recordings are of high quality and clearly demonstrate robust responses to the NCS. Overall, the findings are novel because response properties of visual cortical neurons to such stimuli have not been previously reported, and may provide clues into how illusory stimuli generate percepts similar to real stimuli. However, there are important concerns about the interpretation that neurons are responding to illusory luminance changes when the NCS contains real luminance changes that are not controlled for. Further, the analysis and framing of the NCS response in terms of top-down inputs / visual hierarchy needs to be clarified and strengthened.

Main concerns

1. The study used stimuli that produce illusory percepts in humans, yet records responses from mouse visual cortex and attempts to link these response properties to illusory percepts. It is unknown whether mice experience similar illusory percepts, especially given the vast difference in visual system across species (including acuity). It is entirely possible that the responses recorded here are unrelated to the illusory percept experienced by humans. Providing behavioral evidence that mice respond to the NCS in a similar way to the LDG would allow the study to link neurophysiological findings more strongly with perception, a claim that is not supported at present by the data. We recognize that this would require new data, but perhaps there is a straightforward way of adding this important (in our view) evidence to conclusively link the neural responses to an illusory percept in a mouse model (a central message of the manuscript). These behavioral measures could range from measures of visual reflexive responses to the NCS (such as OKR reflex), or providing a behavioral readout (approach, avoidance, freezing, fleeing, etc.)

contingent upon perception of the orientation / direction of the NCS stimulus (versus control LDG stimuli). Absent this data, there is no definitive evidence to support claims of an illusory “percept” and such claims need to be significantly modified / dialed back in the title and text.

2. Aside from the claim of an “illusory” percept, the study should more carefully control for luminance changes that might be responsible for the responses to the NCS stimulus. Responses to NCS are interpreted as evidence that cells are responding to illusory luminance changes. However, the NCS stimulus contains physical luminance changes within the concentric rings and the reported NCS responses could be driven by these luminance changes occurring within the cell’s receptive field. Such responses would have similar orientation and direction tuning as responses to LDG because the NCS luminance changes are still directional. Furthermore, the F1 modulation would also be 180 degrees out of phase (compared to LDG gratings) because in the NCS, as the white circles become gray, they reduce the mean luminance within the circles whereas in the LDG, as the gray bar sweeps across the circle, it increases the mean luminance within the circles (Fig2j). Finally, the control DBC stimulus would produce a weak response because the relative luminance change is much smaller as the gray moves across the concentric circles due to the surrounding thick white blockers. Therefore, the concern is that responses to luminance changes within the cell’s receptive field could explain many of the properties claimed to be responses to illusory brightness (a central conclusion of the study). To address this, it should be shown that luminance changes within the receptive field are not the main drivers of NCS responses. One solution would be to record the responses to the NCS and test the LDG at multiple different luminance levels to mimic the mean luminance changes in the circles during the NCS. If the NCS response is enhanced compared to the response to LDG at a “matched” overall luminance level, then this would provide further evidence that the NCS response is more than that expected by the physical luminance changes in the NCS stimulus. A second solution would be to identify cells whose RF is centered in a region between the concentric rings, so that there is no luminance change in the RF, but there are responses to the NCS (with comparison to the features of the LDG stimulus).

3. A related concern is that the receptive field location of the recorded cells relative to the grid of concentric circles is not described. Thus claims about illusory brightness response are difficult to interpret since it is unknown whether a cell’s receptive field overlaps the region of concentric circles where physical luminance changes occur in the NCS. Since the experiments also used flashed squares to map the receptive field to center the grating stimuli on the monitor, it would be revealing to analyze the NCS responses relative to the cell’s receptive field location. Cells with receptive fields overlapping the concentric circles would be most susceptible to responding to the physical luminance changes in the NCS whereas cells with receptive fields in the “dark” regions should not. If the authors can demonstrate that these cells exhibit responses to the NCS this would greatly bolster the claim that cells respond to illusory brightness. The hope is that this analysis can help provide a distinction between cells whose NCS response can be explained by physical luminance changes and those that cannot.

4. Similarly, one could imagine that if NCS responses are generated by physical luminance changes in the receptive field, then the cells with strong surround suppression may experience less surround

suppression during the NCS stimulus compared to LDG because the luminance changes are less spatially extended. Is there a relationship between the RF location of a neuron (relative to the concentric rings), and the degree of surround suppression to the LDG? And does this correlate with relatively greater responsiveness to the NCS?

5. The relationship of NCS responses to visual hierarchy underlies the claim that NCS responses are supplied by top-down inputs—however there are no data that directly implicate top-down inputs. If the study could show that manipulation of top-down inputs or upstream areas has a specific effect on NCS responses, this would provide a major advance in understanding the source and mechanism of NCS responses. Barring that evidence, strong claims about the top-down origins of NCS information should be scaled back.

6. Related to the above, it is suggested that processing of NCS stimuli occurs at higher levels of the visual hierarchy involving horizontal / vertical connections in V1, and these interpretations are more grounded in several lines of evidence: long neural response latency to NCS stimuli, greater NCS preference for surround-suppressed cells, greater NCS F1 preference in complex cells, and greater NCS preference in putative interneurons. These data are supportive of NCS responses involving later stage processing in the hierarchy, but because of the aforementioned concerns about physical luminance changes, these analyses need to be re-assessed after addressing those concerns to ensure that the correlations of NCS preference to other functional properties are not just a consequence of physical luminance changes. For example, the longer latency in NCS response vs LDG response (Fig3a) could be explained by the fact that greater physical luminance of the LDG stimulus drives larger and faster feedforward inputs compared to weaker physical luminance of the NCS rather than greater number of serial synaptic activations. Regardless, this analysis should be also be restricted to subsets of cells where NCS and LDG responses have similar magnitudes to ensure that the differences in latency are not due to correlations between latency and response size.

7. Further, the evidence for NCS responses arising further down the V1 hierarchy could be enhanced by adding additional analyses related to cortical layers to make it a more comprehensive analysis of the NCS responses in the context of V1 organization. For example, since the study used linear electrode arrays, can laminar differences in the prevalence / strength of NCS responses be segregated? Do these correlate with latencies? Cells in layer 4 would be expected to have less NCS response whereas “higher” L2/3 and L5 cells are expected to have more. Layer assignment could be done by computing visually-evoked CSD across the linear probe (Niel and Stryker, 2008) or by histologically reconstructing the probe sites. Cells in different layers of mouse visual cortex are known to be targeted by different sources of input (Young et al, eLife 2021; Ji et al. Neuron 2015) so if there are certain patterns of NCS preference across layers, it provides further information as to what circuits are involved.

8. Related to the above, to expand the claims that NCS-preferring neurons are higher in the visual hierarchy, the following measures should be analyzed further with respect to NCS preference: Response

latency to the static receptive field mapping, to provide a general measure of visual response latency for each cell which increases along the hierarchy (this will include the simple cells, an important aspect to clarify); Receptive field size, which also increases along the hierarchy and may be layer-specific.

9. The finding that narrow-spiking putative interneurons are more responsive to NCS than regular-spiking excitatory neurons is interpreted as more evidence of NCS responses being processed at later stages in the hierarchy. However, the narrow-spiking putative interneurons are likely to be PV+ fast-spiking cells (Pfeffer et al. Nature Neuroscience 2013) that are known to be major recipients of feedforward excitation including from dLGN (Kloc and Maffei J Neurosci 2013). Their position in the visual hierarchy likely depends on the layer of origin and other factors. They are also quite visually responsive and may respond to lower contrast/luminance stimuli compared to excitatory neurons (Zhuang et al. JNeurosci 2013) which may explain their stronger NCS responses if their receptive fields overlap with the regions of physical luminance change. Again this analysis needs to be re-assessed after addressing concerns regarding physical luminance responses. The analysis of putative excitatory and inhibitory neurons is interesting and should be included, but the claim that this is also evidence of involvement of top-down feedback should be scaled back.

Minor Concerns:

1. Fig. 1 – it would be very helpful to provide some scale bars for the stimulus display and adjacent comparison to the RF size / visual angle subtended during the experiments
2. Fig. 1B – How is the Perceptual luminance determined?
3. Fig. 4 – please provide responses to LDG in addition to IGR for cell types, for comparison

Below, we give a point-by-point response to the Reviewers. The reviewer's comments are in blue text, and our response is in black text. Changes in the manuscript are highlighted in yellow, except where changes were extensive (title and abstract).

Reviewer #1

The manuscript by Saeedi and colleagues describes the neuronal correlates of illusory brightness in V1. The aim of the manuscript is valuable, as it complements the current literature on perceptual illusions in rodents, a model that can help establish the neuron-level mechanisms of illusory percepts. The presented analyses are generally well done, but I have some concerns, mainly about the statistical approach and the conclusions of the authors.

Reviewer 1, Major Comment #1

The authors claim that the fact that information about illusory brightness is encoded in V1 is evidence that the illusion works as well in mice as in humans. However, in the absence of behavioral evidence, this claim remains, in my opinion, a bit speculative. I would recommend that the authors carefully check the wording of their main claim and add a discussion point about the lack of behavioral confirmation of the perceptual illusion.

We agree with the Reviewer that the main claim should be tempered without behavioral evidence. We addressed the Reviewer's point in two ways. First, we added a behavioral measure: the pupillary response (see lines 470-508 and new Figure 6 in the results). This measure was suggested by Reviewer #3, and we followed their suggestion. We summarize these new experiments and analyses briefly as follows.

Subjective perception of brightness or darkness evokes a pupillary response in humans (Binda, Pereverzeva, & Murray, 2013; Laeng & Endestad, 2012; Naber & Nakayama, 2013; Zavagno, Tommasi, & Laeng, 2017). Moreover, brightness illusions that cause pupil constriction in humans also do so in rats (Vasilev, Raposo, & Totah, 2022). Therefore, the pupillary response can be used as *putative and indirect* evidence of perception. We hypothesized that the pupil would dilate after the NCS stimulus because it is perceived by humans to have "darker than black" gratings. On the other hand, we predicted the opposite pupil response (constriction) after the LDG stimulus, which has the opposite perceived brightness change. As a control, we expected the DBC stimulus not to evoke a PLR because it is physically identical to the NCS stimulus but contains occluders, which block the perception of darkness in humans. Our new experiments (6 mice) are in line with these predictions.

However, we emphasize that this behavioral measure is indirect and write that this is "a reflexive behavioral report that indirectly supports the perception of the illusory brightness by mice, as it does in humans." In the discussion section, we recognize that these new behavioral experiments provide only indirect support of perception, and we discuss that future experiments are needed to link the neuronal activity in mice with a direct behavioral report.

While our findings do not directly support perception in mice, they do unravel, at a new level of detail, the cellular activity underlying sensory processing of illusory brightness across the visual cortical hierarchy. Please see the discussion on lines 592–604.

Reviewer 1, Major Comment #2

The data originates from 32 recordings in 6 mice (if I interpreted the methods correctly). However, the statistical analyses do not take this into account but consider each neuron as an independent unit from the statistical point of view. The authors should rather perform multi-level statistics, taking into account the nested nature of their data.

We appreciate the suggestion of the Review to use statistics that consider the nested nature of the data. We have performed a linear mixed effects model (LME) analysis to address this concern. In our LME model, we incorporated two levels of hierarchy: the mouse ID to account for variability among individual mice and the recording days to account for variability across different days of data collection. This approach allows us to model both the fixed effects of our measurements and the random variability introduced by individual mice and recording days, thus ensuring that our statistical analyses are robust and account for the nested nature of our data. *Importantly, all findings remain supported.*

Reviewer 1, Minor Comment #1

In view of the large number of recorded units, it would be interesting if the authors could estimate whether effects vary across cortical layers. This would be especially relevant to validate the claim that encoding of illusory brightness originates from top-down modulation.

The Reviewer's suggestion to assess laminar responses in support of top-down modulation was helpful. We added a current source density (CSD) analysis of the local field potentials, allowing us to estimate each single unit's cortical layer. We compared units' illusory grating response (IGR) from different layers. We found that units in layer six have weaker responses to the NCS stimulus. There was no significant difference in IGR between other layers. We did not observe laminar differences in the response latencies. We report these new results on lines 406-421 and Supplementary Fig. 12.

We conclude that V1 units that respond to illusory brightness have a bias toward the superficial layers, which receive top-down cortico-cortical feedback, and the granular layer receiving thalamo-cortical input. These results do not directly support a role for top-down modulation in V1 processing of illusory brightness. However, they are an important contribution to understanding what features of the V1 microcircuitry are responsive to illusory brightness, which (as the Reviewer has noted), “help[s] establish the neuron-level mechanisms [of illusory brightness].”

Although we could not establish a clear answer regarding top-down modulation using a laminar analysis, we conducted new experiments in which we optogenetically inhibited higher visual areas (HVAs) that provide feedback to V1, while recording the V1 single unit responses to the NCS, LDG, and DBC stimuli. These experiments demonstrated that inhibition of HVAs diminishes the V1 single unit response *specifically to NCS stimuli*, but not LDG or DBC stimuli. Our findings show that HVA activity modulates the V1 response to illusory brightness. This finding provides support for the role of top-down feedback in processing of illusory brightness by V1 single units. These new results are reported on lines 423-448 and Figure 5 of the results, and our interpretation of them is discussed on lines 576-588 of the discussion section.

Reviewer 1, Minor Comment #2

Mice were sitting on a disc, which I assume could be moved by animals when they attempted locomotion. If this is the case, did the authors assess when animals were moving (i.e. in a state of high arousal) and did they exclude such epochs? It would be important at least to better specify the experimental setup and, if possible, separately analyze periods of motion, in view of the large effect of arousal on sensory processing.

Awake mice were free to run or remain immobile while head-fixed on a “disk”. We added this statement early in the results section (see line 54) and the methods section (see line 607).

As suggested by the Reviewer, we added new data from separate experiments (N=6 mice), in which we simultaneously recorded running speed and V1 single-unit responses to the NCS and control stimuli. Although running was associated with increased stimulus-evoked response magnitude (in line with prior work (Dadarlat & Stryker, 2017; Niell & Stryker, 2010)), the modulation was similar for the NCS and control stimuli. Additionally, running was not associated with a change in the selectivity of V1 units for the NCS or control stimuli. We present these findings on lines 125-137 and in a new Supplementary Fig. 3. These data show that differences in locomotion do not explain our findings about the V1 response to illusory brightness.

References in response to Reviewer 1# comments

- Adelson, E. H., & Bergen, J. R. (1985). Spatiotemporal energy models for the perception of motion. *Journal of Vision*, 2(2), 284-299.
- Alonso, J.-M., & Martinez, L. M. (1998). Functional connectivity between simple cells and complex cells in cat striate cortex. *Nature neuroscience*, 1(5), 395-403.
- Binda, P., Pereverzeva, M., & Murray, S. O. (2013). Pupil constrictions to photographs of the sun. *Journal of vision*, 13(6), 8-8.
- Charbonneau, J. A., Maister, L., Tsakiris, M., & Bliss-Moreau, E. (2022). Rhesus monkeys have an interoceptive sense of their beating hearts. *Proceedings of the National Academy of Sciences*, 119(16), e2119868119.
- Gilbert, C. D., & Wiesel, T. N. (1979). Morphology and intracortical projections of functionally characterised neurones in the cat visual cortex. *Nature*, 280(5718), 120-125.
- Laeng, B., & Endestad, T. (2012). Bright illusions reduce the eye's pupil. *Proceedings of the National Academy of Sciences*, 109(6), 2162-2167.
- Martin, K., & Whitteridge, D. (1984). Form, function and intracortical projections of spiny neurones in the striate visual cortex of the cat. *The Journal of physiology*, 353(1), 463-504.
- Movshon, J. A., Thompson, I. D., & Tolhurst, D. J. (1978). Receptive field organization of complex cells in the cat's striate cortex. *The Journal of physiology*, 283(1), 79-99.
- Naber, M., & Nakayama, K. (2013). Pupil responses to high-level image content. *Journal of vision*, 13(6), 7-7.
- Saravanan, V., Berman, G. J., & Sober, S. J. (2020). Application of the hierarchical bootstrap to multi-level data in neuroscience. *Neurons, behavior, data analysis and theory*, 3(5).
- Tafazoli, S., Safaai, H., De Franceschi, G., Rosselli, F. B., Vanzella, W., Riggi, M., . . . Zoccolan, D. (2017). Emergence of transformation-tolerant representations of visual objects in rat lateral extrastriate cortex. *Elife*, 6, e22794.

- Vasilev, D., Raposo, I., & Totah, N. (2022). Brightness illusions evoke pupil constriction and a visual cortex response in rats.
- Vasilev, D., Raposo, I., & Totah, N. K. (2022). Brightness illusions evoke pupil constriction and a preceding primary visual cortex response in rats. *bioRxiv*, 2022.2007. 2013.499566.
- Zavagno, D., Tommasi, L., & Laeng, B. (2017). The eye pupil's response to static and dynamic illusions of luminosity and darkness. *i-Perception*, 8(4), 2041669517717754.

Reviewer #2

The manuscript by Saeedi and colleagues titled ‘Mouse primary visual cortex neurons respond to the illusory “darker than black” in neon color spreading’ examines whether the primary visual cortex (V1) in mouse contains neurons that respond to the illusion of drifting gratings. Specifically, the illusion used here is referred to as ‘neon’ in the human literature. The development of the ‘neon’ stimulus is novel and exciting. The authors used this stimulus to convincingly demonstrate that mouse neurons in V1 respond to the illusion as if there was a grating present. The characterization of responses to classic gratings and the illusion for the same neurons is informative, and the discovery that the temporal dynamics of the illusion response is delayed is of interest because it implies that circuit elements are ‘filling-in’ a feature not present in the actual stimulus. To be of broad interest, it would be useful if the authors could provide further evidence testing their hypothesis that top-down inputs contribute to the illusion responses, or alternatively demonstrate that top-down inputs do not contribute- either outcome would be of interest, and to pin-point the layers in which the illusion response is observed most prominently. Overall, although the new stimulus is novel and has potential to lead to interesting discoveries, it is unclear how this study advances our understanding of visual processing.

We appreciate the Reviewer’s constructive comments, and their enthusiasm for our paradigm and findings. In response to the Reviewer’s comments, we conducted extensive new analyses and experiments that directly address the need for further evidence to test our hypothesis about top-down inputs.

We have also carefully thought about the Reviewer’s question about how this study advances our understanding of visual processing. In our view, this work advances our understanding of visual processing. First, we demonstrate a V1 single unit response to illusory brightness, which has been lacking. Second, we report the delayed temporal dynamics of the illusion response and its susceptibility to top-down modulation (see new data, discussed below), which tests a major theory for perceptual processing (Gilbert & Li, 2013). Finally, we *characterize the functional features of the V1 microcircuit* that respond to illusory brightness. These features were unknown and included surround suppression, complex cell activation, a preference for superficial and granular layers, and preferential activation of interneurons. The characterization of the V1 microcircuit is a major contribution to understanding visual processing.

Perhaps most importantly, we advance the understanding of visual processing with new laminar analysis and new optogenetics experiments that demonstrate a role for higher visual areas (HVAs) in the V1 single unit response to illusory brightness.

We added a current source density (CSD) analysis of the local field potentials, allowing us to estimate each single unit's cortical layer. We compared units' illusory grating response (IGR) from different layers. We found that units in layer six have weaker responses to the NCS stimulus. There was no significant difference in IGR between other layers. We did not observe laminar differences in the response latencies. We report these new results on lines 406-421 and Supplementary Fig. 12.

We conclude that V1 units that respond to illusory brightness have a bias toward the superficial layers, which receive top-down cortico-cortical feedback, and the granular layer receiving

thalamo-cortical input. These results do not directly support a role for top-down modulation in V1 processing of illusory brightness. However, they are an important contribution to understanding what features of the V1 microcircuitry are responsive to illusory brightness.

Although we could not establish a clear answer regarding top-down modulation using a laminar analysis, we conducted new experiments in which we optogenetically inhibited higher visual areas (HVAs) that provide feedback to V1, while recording the V1 single unit responses to the NCS, LDG, and DBC stimuli. These experiments demonstrated that inhibition of HVAs diminishes the V1 single unit response *specifically to NCS stimuli*, but not LDG or DBC stimuli. Our findings show that HVA activity modulates the V1 response to illusory brightness. This finding provides support for the role of top-down feedback in processing of illusory brightness by V1 single units. These new results are reported on lines 423-448 and Figure 5 of the results, and our interpretation of them is discussed on lines 576-588 of the discussion section.

Reviewer 2, Major comment #1

Significance of the manuscript would be improved if the authors identified the level in the visual hierarchy in which responses to the illusion first appeared. For example, are the layer 2/3 responses inherited from layer 4 or is layer 2/3 the first site of emergence?

We agree with the Reviewer. Please see the above response, in which we discuss the new laminar analysis added to the manuscript.

Reviewer 2, Major comment #2

Do the authors have evidence disentangling the two possible interpretations, intra-V1 versus extra-V1? For example, if V2 is inactivated do the responses to the illusion weaken?

We have added new experiments to answer the question posed by the Reviewer. Please see the discussion of these in the first response to Reviewer 2 (above).

Reviewer 2, Major comment #3

The extent to which complex cells in the mouse preferentially receive top-down feedback is unknown; therefore, the authors have not met the bar to claim that their data indicate top-down input plays a role in the observed response to the illusion stimulus. What is the evidence in mouse that complex neurons preferentially receive top-down inputs over for example, simple cells?

We agree with the Reviewer. We have completely removed the speculation that the activation of complex cells is evidence for top-down input into V1. Instead, we only discuss the findings regarding complex cells in the context of characterizing the features of the V1 microcircuit that respond to illusory brightness. Please see lines 356 and 357 of the results section and lines 561-564 of the discussion section for the relevant changes.

We now support the role of top-down inputs with optogenetics inhibition of the HVAs (see discussion above), and we *only discuss top-down inputs in the context of those optogenetics inhibition experiments*.

Reviewer 2, Major comment #4

The discussion on the role of interneurons is opened-ended, and vague. Does the authors' argument require that interneurons are preferentially contacted by top-down feedback over specific classes of excitatory neurons? If so, please provide evidence, either experimentally or in literature.

We agree with the Reviewer. As we have done for complex cells, we have completely removed the speculation that interneuron activity is a sign of top-down input into V1. We only discuss the findings regarding putative interneurons in the context of characterizing the features of the V1 microcircuit that respond to illusory brightness. We no longer make any statements about the function of the interneurons. We discuss the role of V1 (putative) interneurons in visual stimulus processing in general and we conclude that V1 interneurons are involved in processing brightness illusions without speculating further. Please see lines 377-380 of the results section and lines 542-546 and 570-575 of the discussion section for the relevant changes.

Reviewer 2, Major comment #5

The rationale for considering preference for NCS versus LDG in Figure 3c is unclear.

We have classified neurons based on their response to NCS stimuli compared to LDG stimuli. Single units with larger responses to NCS were labeled as NCS-preferring, and those with larger responses to LDG were labeled as LDG-preferring neurons. With such a grouping, we could statistically test distributions of NCS-preferring and LDG-preferring units have significant differences in terms of susceptibility to surround suppression.

Due to extensive new experiments and analyses that were added to the manuscript, the relevant Figure is now Fig. 4d. Accordingly, we have revised the legend in Fig. 4 and clarified the main text (see lines 322-323)

Reviewer 2, Major comment #6

It is unclear how narrow-spiking neurons relate to surround modulation, given it has been shown that somatostatin neurons, which generally have a regular-shaped waveform, contribute to surround modulation.

We agree with the Reviewer's statement that the relation of narrow spiking neurons to surround modulation is unclear. We have removed this speculation. We have confined our discussion of putative interneurons to simply state that they are responsive to illusory brightness stimuli.

Reviewer 2, Major comment #7

If the recordings were done in awake animals, please make that clear. Given the hypothesis is that top-down feedback contributes to the illusion responses, it would be problematic if the recording were done in an anesthetized prep.

The recordings were performed on awake head-fixed mice. There is no anesthetized (or even "quiet wakefulness") data in this manuscript. We appreciate the Reviewer drawing our

attention to this aspect of the experiment being unclear. It is now stated clearly at the start of the results (see line 54) and clarified in the methods (see line 607).

Reviewer 2, Minor comment #1

Title could be improved to capture a broader audience. As written, I think only visual neuroscientists will relate. The use of 'color' in the title and discussion is a little confusing. Although inspired by color spreading studies in humans, there is no color used here.

We thank the Reviewer for this helpful insight. We changed the title to **“Brightness illusions drive a neuronal response in the primary visual cortex under top-down modulation.”** This new title is more general and captures the main new findings here, which are that V1 single units respond to brightness illusions, when prior work had been limited to human fMRI rather than single cells. It also highlights the new experiments demonstrating a role for modulation of the V1 single unit response by higher visual areas. We sincerely appreciate this feedback, which has improved the clarity of the title and how it communicates the impact to a broader audience.

We also agree with the Reviewer that the use of color / “neon” is confusing. We have modified this throughout the manuscript. We state that we are using an achromatic version of the neon color spreading illusion (see lines 37, 67, 525).

Reviewer 2, Minor comment #2

Depth of recoding from pia surface should be reported.

We have reported the recording depth in Supplementary Table 1 and referred to it in the Methods section, line 635.

Reviewer 2, Minor comment #3

More details on the receptive field mapping should be incorporated into the analysis of the illusion responses, to ensure that the main effect holds if only neurons that were stimulated in the center of their receptive field are included in analysis.

We have addressed the Reviewer’s concern by performing the suggested analysis. In a new set of experiments, we used full-screen stimuli with concentric circles arranged on a honeycomb structure providing maximal distance and illusory areas between the concentric circles. We recorded 234 V1 single units from 13 mice in these new experiments. The RF of these units had no overlap with concentric circles (inducers) and they were not responsive to the DBC stimulus, which means that they did not respond to the physically identical stimulus that differed only in that the illusion was blocked. We replicated our main finding with this new set of units. These new results are presented in Fig. 3 and lines 196-229.

Reviewer 2, Minor comment #4:

Online site for code availability is missing.

The code will be uploaded to GitHub upon acceptance.

Reviewer 2, Minor comment #5:

More examples of stimulus responses would be useful.

We have added additional examples. Please see Supplementary Fig. 2.

Reviewer 2, Minor comment #6:

The number of animals used in each plot is unclear.

We have added the number of animals to the legend of every figure. We have also added the number to the main text of the results section.

Reviewer 2, Minor comment #7

The density of simple cells differs across layers, please account for this in the analysis and interpretation.

As we mentioned in the response to Reviewer 2 (Major comment #1), we have performed the CSD analysis and identified the cortical layer in which neuron belong to. Although the laminar density of simple/complex cells is not constant, we did not find a laminar difference between their IGR index or NCS delay index (see the below figure).

References in response to Reviewer #2 comments

- Binda, P., Pereverzeva, M., & Murray, S. O. (2013). Pupil constrictions to photographs of the sun. *Journal of vision*, 13(6), 8-8.
- Dadarlat, M. C., & Stryker, M. P. (2017). Locomotion enhances neural encoding of visual stimuli in mouse V1. *Journal of Neuroscience*, 37(14), 3764-3775.
- Gilbert, C. D., & Li, W. (2013). Top-down influences on visual processing. *Nature Reviews Neuroscience*, 14(5), 350-363.
- Laeng, B., & Endestad, T. (2012). Bright illusions reduce the eye's pupil. *Proceedings of the National Academy of Sciences*, 109(6), 2162-2167.
- Naber, M., & Nakayama, K. (2013). Pupil responses to high-level image content. *Journal of vision*, 13(6), 7-7.
- Niell, C. M., & Stryker, M. P. (2010). Modulation of visual responses by behavioral state in mouse visual cortex. *Neuron*, 65(4), 472-479.
- Vasilev, D., Raposo, I., & Totah, N. (2022). Brightness illusions evoke pupil constriction and a visual cortex response in rats.
- Zavagno, D., Tommasi, L., & Laeng, B. (2017). The eye pupil's response to static and dynamic illusions of luminosity and darkness. *i-Perception*, 8(4), 2041669517717754.

Reviewer #3

The manuscript by Saeedi et al. demonstrates that neurons in mouse visual cortex respond to a visual stimulus (neon color spreading stimulus, NCS) that induces perception of an illusory drifting grating in humans. The study reports that NCS-responding neurons have similar orientation and direction tuning to NCS and real luminance-defined gratings (LDG) stimuli, and have F1 responses that correspond to the spatial phase features of each stimulus. Neural preferences for NCS vs LDG correlate with other properties such as surround suppression, relative F1 vs F0 response (simple/complex), and spike waveforms. NCS-preferring neurons also exhibit greater surround suppression and have more complex-cell-like properties, and cells with fast-spiking waveforms had greater preference for NCS. From these neural responses, the study puts forth two main conclusions: 1) mouse visual cortical neurons are responding to NCS stimuli and thus respond to illusory luminance changes (as perceived by humans) and 2) NCS-preferring neurons are positioned higher in the visual hierarchy, allowing them to receive more top-down inputs which could contribute to their enhanced responses to NCS stimuli. The recordings are of high quality and clearly demonstrate robust responses to the NCS. Overall, the findings are novel because response properties of visual cortical neurons to such stimuli have not been previously reported, and may provide clues into how illusory stimuli generate percepts similar to real stimuli. However, there are important concerns about the interpretation that neurons are responding to illusory luminance changes when the NCS contains real luminance changes that are not controlled for. Further, the analysis and framing of the NCS response in terms of top-down inputs / visual hierarchy needs to be clarified and strengthened.

We appreciate the reviewer's thoughtful evaluation of our manuscript and their positive remarks regarding the novelty and quality of our findings. In the following, we address their concerns by further clarifying the writing and adding extensive data from new experiments.

Reviewer 3, Major comment #1

The study used stimuli that produce illusory percepts in humans, yet records responses from mouse visual cortex and attempts to link these response properties to illusory percepts. It is unknown whether mice experience similar illusory percepts, especially given the vast difference in visual system across species (including acuity). It is entirely possible that the responses recorded here are unrelated to the illusory percept experienced by humans. Providing behavioral evidence that mice respond to the NCS in a similar way to the LDG would allow the study to link neurophysiological findings more strongly with perception, a claim that is not supported at present by the data. We recognize that this would require new data, but perhaps there is a straightforward way of adding this important (in our view) evidence to conclusively link the neural responses to an illusory percept in a mouse model (a central message of the manuscript). These behavioral measures could range from measures of visual reflexive responses to the NCS (such as OKR reflex), or providing a behavioral readout (approach, avoidance, freezing, fleeing, etc.) contingent upon perception of the orientation / direction of the NCS stimulus (versus control LDG stimuli). Absent this data, there is no definitive evidence to support claims of an illusory “percept” and such claims need to be significantly modified/dialed back in the title and text.

We agree with the Reviewer that the main claim should be tempered without behavioral evidence. We addressed the Reviewer’s point in two ways. First, we added a behavioral

measure: the pupillary response (see lines 470-508 and new Figure 6 in the results). Second, we narrow the scope of our claim according to available evidence and discuss the limitations in our experiments (see lines 592 –604 in discussion).

We summarize these new experiments and analyses briefly as follows.

Subjective perception of brightness or darkness evokes a pupillary response in humans (Binda, Pereverzeva, & Murray, 2013; Laeng & Endestad, 2012; Naber & Nakayama, 2013; Zavagno, Tommasi, & Laeng, 2017). Moreover, brightness illusions that cause pupil constriction in humans also do so in rats (Vasilev, Raposo, & Totah, 2022). Therefore, the pupillary response can be used as *putative and indirect* evidence of perception. We hypothesized that the pupil would dilate after the NCS stimulus because it is perceived by humans to have “darker than black” gratings. On the other hand, we predicted the opposite pupil response (constriction) after the LDG stimulus, which has the opposite perceived brightness change. As a control, we expected the DBC stimulus not to evoke a PLR because it is physically identical to the NCS stimulus but contained occluders, which block the perception of darkness in humans. Our new experiments (6 mice) are in line with these predictions.

However, we emphasize that this behavioral measure is indirect and write that this is “a reflexive behavioral report that indirectly supports the perception of the illusory brightness by mice, as it does in humans.” In the discussion section, we recognize that these new behavioral experiments provide only indirect support of perception, and we discuss that future experiments are needed to link the neuronal activity in mice with a direct behavioral report.

While our findings do not directly support perception in mice, they do unravel, at a new level of detail, the cellular activity underlying sensory processing of illusory brightness across the visual cortical hierarchy. Please see the discussion on lines 592 –604.

We also appreciate the Reviewer’s suggestion to use the OKR reflex. We found one paper on the rodent OKR (Tabata, Shimizu, Wada, Miura, & Kawano, 2010). Based on that work, there were a number of issues that precluded us from examining the OKR. First, the duration that we presented the visual stimuli is not sufficient for estimating the full content of the OKR according to Tabata, et al. (2010). Instead, we were limited to examining only the initial segment of the OKR. However, according to Tabata, et al. (2010), analysis of this initial segment requires either a higher spatial frequency than our stimuli or a lower temporal frequency than our study. Second, the gratings covered the whole visual field while in our experiments, while in Tabata, et al. (2010) the stimulus was presented to only one eye. This results in less OKR.

We also considered the Reviewer’s recommendation to use approach, avoidance, freezing, fleeing behaviors. These are challenging to implement in the head-fixed preparation. Therefore, we used the pupillary response, which is supported by human and rat studies. We recognize that this is indirect evidence and we discuss the need for direct evidence from behavioral report such as a brightness discrimination task. This work can now be pursued since our work identifies that this illusion is associated with a pupil response and V1 single unit response in mice.

Reviewer 3, Major comment #2

Aside from the claim of an “illusory” percept, the study should more carefully control for luminance changes that might be responsible for the responses to the NCS stimulus. Responses to NCS are interpreted as evidence that cells are responding to illusory luminance changes. However, the NCS stimulus contains physical luminance changes within the concentric rings and the reported NCS responses could be driven by these luminance changes occurring within the cell’s receptive field. Such responses would have similar orientation and direction tuning as responses to LDG because the NCS luminance changes are still directional. Furthermore, the F1 modulation would also be 180 degrees out of phase (compared to LDG gratings) because in the NCS, as the white circles become gray, they reduce the mean luminance within the circles whereas in the LDG, as the gray bar sweeps across the circle, it increases the mean luminance within the circles (Fig2j). Finally, the control DBC stimulus would produce a weak response because the relative luminance change is much smaller as the gray moves across the concentric circles due to the surrounding thick white blockers. Therefore, the concern is that responses to luminance changes within the cell’s receptive field could explain many of the properties claimed to be responses to illusory brightness (a central conclusion of the study). To address this, it should be shown that luminance changes within the receptive field are not the main drivers of NCS responses. One solution would be to record the responses to the NCS and test the LDG at multiple different luminance levels to mimic the mean luminance changes in the circles during the NCS. If the NCS response is enhanced compared to the response to LDG at a “matched” overall luminance level, then this would provide further evidence that the NCS response is more than that expected by the physical luminance changes in the NCS stimulus. A second solution would be to identify cells whose RF is centered in a region between the concentric rings, so that there is no luminance change in the RF, but there are responses to the NCS (with comparison to the features of the LDG stimulus).

We thank the Reviewer for the detailed comment and valuable suggestions. We have implemented the second suggested solution. In a new set of experiments in 13 mice, we used full-screen stimuli with concentric circles arranged on a honeycomb structure providing maximal distance and illusory areas between the concentric circles. We recorded 234 V1 single units with a RF covering only the illusory area. These units were not responsive to the DBC stimulus, which means that they did not respond to the physically identical stimulus that differed only in that the illusion was blocked.

In this new set of units, we replicated the main findings. Firstly, we found that about 50% of these units are responding to the NCS stimuli, which shows that the NCS responses are not due to the direct stimulation within the RF (Fig. 3a-d). Secondly, we observed that the preferred direction is preserved in most of these neurons (Fig. 3e). Finally, we found that the temporal modulation of F1 responses was in the opposite phase for NCS stimuli compared to LDG stimuli (Fig. 3f). This effect remains intact for different subpopulations of single units with different amounts of RF-inducer overlap (please see Supplementary Fig. 5). These new results have been added to the paper (see lines 196-229).

Reviewer 3, Major comment #3

A related concern is that the receptive field location of the recorded cells relative to the grid of concentric circles is not described. Thus, claims about illusory brightness response are difficult to interpret since it is unknown whether a cell’s receptive field overlaps the region of concentric circles where physical luminance changes occur in the NCS. Since the experiments also used

flashed squares to map the receptive field to center the grating stimuli on the monitor, it would be revealing to analyze the NCS responses relative to the cell's receptive field location. Cells with receptive fields overlapping the concentric circles would be most susceptible to responding to the physical luminance changes in the NCS whereas cells with receptive fields in the "dark" regions should not. If the authors can demonstrate that these cells exhibit responses to the NCS this would greatly bolster the claim that cells respond to illusory brightness. The hope is that this analysis can help provide a distinction between cells whose NCS response can be explained by physical luminance changes and those that cannot.

We thank the Reviewer for their critical assessment of our findings and for suggesting this analysis to bolster the claim that cells respond to illusory brightness. In response to the Reviewer's Major comment #2, we excluded single units with a RF overlapping the concentric circles (inducers). In response to Major Comment #3, we look at the IGR responses versus RF-inducer overlap to investigate the effect of small luminance changes in the illusory grating on the single unit response to the NCS stimulus. We found that the IGR index and RF overlap with the inducers are only weakly correlated ($r=0.09$, $p=6.23e-3$). Therefore, we conclude that physical luminance changes in the NCS stimulus are not the primary factor driving the neuronal NCS responses. We report these data in a new figure (Supplementary Fig. 4) and report the correlation in the results (see lines 211-217).

Reviewer 3, Major comment #4

Similarly, one could imagine that if NCS responses are generated by physical luminance changes in the receptive field, then the cells with strong surround suppression may experience less surround suppression during the NCS stimulus compared to LDG because the luminance changes are less spatially extended. Is there a relationship between the RF location of a neuron (relative to the concentric rings), and the degree of surround suppression to the LDG? And does this correlate with relatively greater responsiveness to the NCS?

While we recognize that reporting a relationship between surround suppression, the RF-inducer overlap, and the responsiveness to the NCS stimuli, it was not possible to perform this test. This would require ~400 min of head-fixation for RF mapping along per mouse (in addition to the time required to perform current experiment on brightness illusions). This is not possible due to local ethical guidelines for animal welfare. However, as addressed in our response to the above comments, additional experiments permitted us to conclude that the single unit response to the illusory grating exists in the absence of physical luminance changes in the receptive field. Although not all experiments could be performed, our findings nevertheless provide an important first step in characterizing the V1 single unit response to illusory brightness, the microcircuitry involved, the role of top-down modulation from HVAs, and a demonstration of pupil dilation to an illusion of darkness.

Reviewer 3, Major comment #5

The relationship of NCS responses to visual hierarchy underlies the claim that NCS responses are supplied by top-down inputs—however there are no data that directly implicate top-down inputs. If the study could show that manipulation of top-down inputs or upstream areas has a specific effect on NCS responses, this would provide a major advance in understanding the

source and mechanism of NCS responses. Barring that evidence, strong claims about the top-down origins of NCS information should be scaled back.

We conducted new experiments in which we optogenetically inhibited higher visual areas (HVAs) that provide feedback to V1, while recording the V1 single unit responses to the NCS, LDG, and DBC stimuli. These experiments demonstrated that inhibition of HVAs diminishes the V1 single unit response *specifically to NCS stimuli*, but not LDG or DBC stimuli. Our findings show that HVA activity modulates the V1 response to illusory brightness. This finding provides support for the role of top-down feedback in processing of illusory brightness by V1 single units. These new results are reported on lines 423-448 and Figure 5 of the results, and our interpretation of them is discussed on lines 576-588 of the discussion section.

Reviewer 3, Major comment #6

Related to the above, it is suggested that processing of NCS stimuli occurs at higher levels of the visual hierarchy involving horizontal / vertical connections in V1, and these interpretations are more grounded in several lines of evidence: long neural response latency to NCS stimuli, greater NCS preference for surround-suppressed cells, greater NCS F1 preference in complex cells, and greater NCS preference in putative interneurons. These data are supportive of NCS responses involving later stage processing in the hierarchy, but because of the aforementioned concerns about physical luminance changes, these analyses need to be re-assessed after addressing those concerns to ensure that the correlations of NCS preference to other functional properties are not just a consequence of physical luminance changes. For example, the longer latency in NCS response vs LDG response (Fig3a) could be explained by the fact that greater physical luminance of the LDG stimulus drives larger and faster feedforward inputs compared to weaker physical luminance of the NCS rather than greater number of serial synaptic activations. Regardless, this analysis should be also be restricted to subsets of cells where NCS and LDG responses have similar magnitudes to ensure that the differences in latency are not due to correlations between latency and response size.

We agree with the Reviewer that one possible reason for our findings might be the greater physical luminance of the LDG stimulus, which drives stronger responses. To check this possibility, we extracted a subset of units with similar response magnitudes (equi-responsive cell) to NCS and LDG stimuli. We identified 286 equi-responsive complex cells and found their responses to the NCS stimulus are likewise delayed compared to LDG stimulus. We have added this finding in the manuscript lines 265-273 and Supplementary Fig. 6.

We would also like to mention that for this subset of neurons with similar magnitude of response, the IGR index is mainly zero and therefore we cannot reassess our analysis of surround modulation, simple complex cells, and putative I/E cells as they are based on IGR differences among the neurons.

Reviewer 3, Major comment #7

Further, the evidence for NCS responses arising further down the V1 hierarchy could be enhanced by adding additional analyses related to cortical layers to make it a more comprehensive analysis of the NCS responses in the context of V1 organization. For example, since the study used linear electrode arrays, can laminar differences in the prevalence / strength

of NCS responses be segregated? Do these correlate with latencies? Cells in layer 4 would be expected to have less NCS response whereas “higher” L2/3 and L5 cells are expected to have more. Layer assignment could be done by computing visually-evoked CSD across the linear probe (Niel and Stryker, 2008) or by histologically reconstructing the probe sites. Cells in different layers of mouse visual cortex are known to be targeted by different sources of input (Young et al, eLife 2021; Ji et al. Neuron 2015) so if there are certain patterns of NCS preference across layers, it provides further information as to what circuits are involved.

We agree with the Reviewer that laminar analysis would provide deeper insights into the circuits involved.

We added a current source density (CSD) analysis of the local field potentials, which allowed us to estimate the cortical layer of each single unit. We compared the illusory grating response (IGR) of units from different layers. We found that units in layer six have weaker responses to the NCS stimulus. There was no significant difference in IGR between other layers. We did not observe laminar differences in the response latencies. We report these new results on lines 406-421 and Supplementary Fig. 12.

We conclude that V1 units that respond to illusory brightness have a bias toward the superficial layers, which receive top-down cortico-cortical feedback, and the granular layer receiving thalamo-cortical input. These results do not directly support a role for top-down modulation in V1 processing of illusory brightness. However, they are an important contribution to understanding what features of the V1 microcircuitry are responsive to illusory brightness.

Although the CSD analysis does not conclusively support top-down modulation of V1 (although it is not inconsistent with that interpretation), we now use optogenetics to support that analysis and confine discussion of laminar results to understanding V1 microcircuitry.

Reviewer 3, Major comment #8

Related to the above, to expand the claims that NCS-preferring neurons are higher in the visual hierarchy, the following measures should be analyzed further with respect to NCS preference. Response latency to the static receptive field mapping, to provide a general measure of visual response latency for each cell which increases along the hierarchy (this will include the simple cells, an important aspect to clarify). Receptive field size, which also increases along the hierarchy and may be layer-specific.

We appreciate the Reviewer’s suggestions and performed the analysis they have asked. We calculated the response latency of units to the static receptive field mapping as a general measure of hierarchies. We show some example PSTH in response to contrast changing rectangles presented in the receptive field mapping experiment and the estimated the response latencies in Supplementary Fig. 7. We then plotted the IGR index against this new measure of latency which hereafter, we call “rectangle latency”. We found that IGR is anti-correlated with the rectangle latency. We have shown these new results in Fig. 4b. Although the NCS responses are delayed compared to LDG responses indicating the requirement of extra synaptic interaction for NCS responses, these new results with rectangle latency were unexpected, as one of our main claims was the NCS-preferring neurons are at the higher level of visual hierarchy within V1. Therefore, we scaled back this claim and tempered our wording. We no longer refer to a within-V1 “hierarchy.” Instead, we only discuss our findings in the context of

characterizing the features of V1 microcircuit that respond to brightness illusions (see lines 295-305).

Reviewer 3, Major comment #9

The finding that narrow-spiking putative interneurons are more responsive to NCS than regular-spiking excitatory neurons is interpreted as more evidence of NCS responses being processed at later stages in the hierarchy. However, the narrow-spiking putative interneurons are likely to be PV+ fast-spiking cells (Pfeffer et al. Nature Neuroscience 2013) that are known to be major recipients of feedforward excitation including from dLGN (Kloc and Maffei J Neuroscience 2013). Their position in the visual hierarchy likely depends on the layer of origin and other factors. They are also quite visually responsive and may respond to lower contrast/luminance stimuli compared to excitatory neurons (Zhuang et al. JNeurosci 2013) which may explain their stronger NCS responses if their receptive fields overlap with the regions of physical luminance change. Again, this analysis needs to be re-assessed after addressing concerns regarding physical luminance responses. The analysis of putative excitatory and inhibitory neurons is interesting and should be included, but the claim that this is also evidence of involvement of top-down feedback should be scaled back.

We agree with all of the points raised by the Reviewer and have modified the manuscript accordingly. First, we have removed all references to a within-V1 “hierarchy” and instead discuss these findings in the context of characterizing the aspects of the V1 microcircuit that respond to illusory brightness. Second, we removed the claim that the responsiveness of V1 interneurons indicates a role for top-down feedback. We now only discuss interneuron activity in the context of describing the V1 single unit physiology and features of the V1 microcircuitry that are responsive to illusory brightness. Third, we have re-assessed interneuron responsiveness after controlling for physical luminance (as discussed in response to the Reviewer’s Major Comment #6, please see above for detailed response). Finally, using the new data collected during the presentation of full screen stimuli and oblique insertion angle of the electrode, we calculated the distance of RF center to the center of closest inducer circles. The design of our new experiment provided a population of units with varying RF-inducer distances. We then calculated the correlation of NCS delay index with the RF-inducer distances for putative interneurons and putative pyramidal neurons. Interestingly, we found a positive correlation for putative interneurons only. We presented these results in Fig. 4f-k and lines 395-405.

Reviewer 3, Minor Concern 1:

Fig. 1 – it would be very helpful to provide some scale bars for the stimulus display and adjacent comparison to the RF size / visual angle subtended during the experiments.

We added the scale bars to the Fig. 1. In addition, we added one example RF mapping (please see Fig. 1d).

Reviewer 3, Minor Concern 2:

Fig. 1B – How is the Perceptual luminance determined?

The mentioned figure is a schematic drawing with the purpose of illustration only. We have clarified this in the legend of the figure by stating that the luminance values are arbitrary (please see line 102). We would like to note that obtaining a proper value for perceptual luminance requires a psychophysical experiment in humans, which does not fit into the scope of the current work. Moreover, the evaluation of perceptual luminance in mice requires additional behavioral experiments that we would like to pursue in the future.

Reviewer 3, Minor Concern 3:

Fig. 4 – please provide responses to LDG in addition to IGR for cell types, for comparison

In the new version of manuscript, we have combined former Fig.4 with Fig. 3. In order to avoid overcrowding this figure, we provided the responses to LDG and NCS stimuli for different cell types (i.e. putative I/E cells) in Supplementary Fig. 11.

References in response to Reviewer #3 comments

- Binda, P., Pereverzeva, M., & Murray, S. O. (2013). Pupil constrictions to photographs of the sun. *Journal of vision, 13*(6), 8-8.
- Laeng, B., & Endestad, T. (2012). Bright illusions reduce the eye's pupil. *Proceedings of the National Academy of Sciences, 109*(6), 2162-2167.
- Naber, M., & Nakayama, K. (2013). Pupil responses to high-level image content. *Journal of vision, 13*(6), 7-7.
- Tabata, H., Shimizu, N., Wada, Y., Miura, K., & Kawano, K. (2010). Initiation of the optokinetic response (OKR) in mice. *Journal of vision, 10*(1), 13-13.
- Vasilev, D., Raposo, I., & Totah, N. (2022). Brightness illusions evoke pupil constriction and a visual cortex response in rats.
- Zavagno, D., Tommasi, L., & Laeng, B. (2017). The eye pupil's response to static and dynamic illusions of luminosity and darkness. *i-Perception, 8*(4), 2041669517717754.

REVIEWERS' COMMENTS

Reviewer #1 (Remarks to the Author):

The authors addressed all my comments satisfactorily. I believe that the addition of the pupil response analysis convincingly shows that the illusion the authors presented might have been in fact perceived by mice.

A remaining question relates to the new experiment in which authors inactivated HVAs. The authors show that responses to illusory stimuli are observed in all V1 layers (but less in deep layers), including the thalamorecipient layer 4. When inactivating HVAs, a decrease in responses to illusory stimuli is observed in V1. However, one would expect layer 4 responses to be spared by HVA inactivation. I realize this might not be easy to answer in view of the limited number of illusion-responsive neurons recorded in the HVA inactivation experiment. Nevertheless, it is important to at least discuss the fact that, besides top-down modulation, bottom-up inputs might already encode responses to illusory stimuli in V1, perhaps already in the thalamus or earlier. This is something that should be explicitly addressed in the discussion.

Reviewer #2 (Remarks to the Author):

In their revision the authors of the manuscript titled 'Brightness illusions drive a neuronal response in the primary visual cortex under top-down modulation' addressed most of the concerns raised by the previous review. Importantly, the authors clarified the novelty and significance of their findings. The authors claim that a major finding of their work is that higher visual areas (HVAs) are pivotal in shaping the response properties of individual primary cortex neurons in response to illusory brightness.

The work is significant and advances our understanding of contextual influence on the responses of primary visual cortex neurons. It is exciting that the authors identified a manipulation (silencing HVAs) that differentially impacts the response to the illusion and standard grating stimuli.

However, to fully support the claim noted above, it would be useful if the authors could perform additional analysis to demonstrate more precisely what this pivotal role is, at the level of individual neurons. Specifically, (1) report the change in average response (e.g. Fig. 5f) on a cell by cell basis separately for each of the six animals [assuming that recordings were done with and without silencing in the same recording session] and (2) provide an estimation of the fraction of the HVA silenced during optogenetic stimulation and whether the HVA neurons that are silenced are retinotopically matched to

the recorded V1 neurons - presumably the entire HVA is not silenced or that would likely spill into V1, so providing information about what the manipulation is functionally doing within the HVAs is important.

Reviewer #3 (Remarks to the Author):

The authors have performed substantial new experiments, analysis, and clarifications of text that address most of the major comments and substantially improve the manuscript. There are only a few remaining minor issues that can be addressed with text revisions.

1. The addition of the pupil data is compelling. We would suggest condensing / moving some of the text in lines 473-501 to the introduction to motivate and highlight the finding for the reader earlier in the manuscript. Also, can the authors mention if they checked whether the overall luminance differences explain the timing / magnitude (not polarity) of the response?
2. The new data showing similar NCS response properties for units with RFs that do not overlap the inducer (non-overlapping) strengthens the claim that neurons in V1 are responding to the illusory brightness. However, some details of these experiments are missing from the Methods. Please make sure to include: Description of the new stimuli with wider spacing between inducers; RF mapping protocol (both “square” and “rectangle” mapping stimuli are referenced in the Results, but only squares are mentioned in the Methods). ; analytic methods for quantifying degree of overlap). It would also be great to include the RF centers and/or RF contours of several examples or the population average of all of the “non-overlapping” units on the same plot relative to the inducers (similar to 3a). This would provide a very convincing demonstration for the readers, and could provide a visual anchor for the authors explanation for the relationship to surround suppression that is unable to be addressed here (prior main comment #4).
3. The new experiments demonstrate a role for HVA inputs in specifically driving NCS response. However the responses to NCS and LDG in LM and LI during the optogenetic manipulation are not shown. Please show a figure or report text results that quantify the responses to the NCS and LDG in LM and LI units during the optogenetic manipulation to demonstrate that they are adequately inhibited.
4. It is worth highlighting in text that the “rectangle latency” results along with laminar analysis demonstrate that the NCS responses do not cleanly map onto laminar or latency-based hierarchical organization. Perhaps worth speculating why (esp. in light of optogenetic results showing some sort of hierarchical influence).
5. The new data and analysis show that after controlling for physical luminance, interneurons and excitatory neurons have similar preference for NCS. However in line 403 the authors suggest that the results demonstrate a role of interneurons in “spatial spreading and filling-in effect” which is not very well explained. How could inhibitory neurons (that are suppressing the transmission of excitatory signals beyond V1) accomplish this?

6. The claim that complex cells contribute more to NCS processing is referenced several times (Line 379, 568) but the IGR shows no correlation with CSM index (Supp Fig. 10), demonstrating that simple and complex cells have similar preferences for NCS vs LDG. IGR_F1 is correlated to CSM, but this does not mean that complex cells are more strongly involved in NCS processing than simple cells. For example, this effect could be due to the low levels of LDG F1 modulation rather than NCS F1 modulation. Therefore, the language regarding the preferential processing of NCS by complex cells (Line 379, 417, 568) should be tempered.

We appreciate your and the Reviewers' comments. These have led to an improved manuscript.

Below, we give a point-by-point response to the Reviewers. The reviewer's comments are in blue text, and our response is in black text. Changes in the manuscript are highlighted in yellow, except where changes were extensive (abstract).

Reviewer #1 (Remarks to the Author):

The authors addressed all my comments satisfactorily. I believe that the addition of the pupil response analysis convincingly shows that the illusion the authors presented might have been, in fact, perceived by mice.

A remaining question relates to the new experiment in which authors inactivated HVAs. The authors show that responses to illusory stimuli are observed in all V1 layers (but less in deep layers), including the thalamorecipient layer 4. When inactivating HVAs, a decrease in responses to illusory stimuli is observed in V1. However, one would expect layer 4 responses to be spared by HVA inactivation. I realize this might not be easy to answer in view of the limited number of illusion-responsive neurons recorded in the HVA inactivation experiment. Nevertheless, it is important to at least discuss the fact that, besides top-down modulation, bottom-up inputs might already encode responses to illusory stimuli in V1, perhaps already in the thalamus or earlier. This is something that should be explicitly addressed in the discussion.

We thank the Reviewer for their constructive comments and for acknowledging our efforts to address their previous concerns.

We agree with the Reviewer that bottom-up inputs may encode responses to illusory stimuli in V1. In addition to the low number of available NCS-responsive neurons noted by the Reviewer, we would add that the oblique angle of electrode insertion required to reach the HVAs precludes the possibility of accurately estimating layer boundaries using current source density (CSD) analysis in the HVA inactivation experiments. We have explicitly addressed this issue and the methodological constraint in the Discussion section of our revised manuscript (lines 623-638).

While acknowledging the limitations of our study in conclusively determining the extent to which bottom-up processes contribute to the observed responses in layer 4 of V1, we suggest that future studies focus on directly recording from thalamic neurons while presenting illusory stimuli, in combination with HVA inactivation. This would help delineate the contributions of thalamic inputs to encoding illusory stimuli in V1.

Reviewer #2 (Remarks to the Author):

In their revision, the authors of the manuscript titled 'Brightness illusions drive a neuronal response in the primary visual cortex under top-down modulation' addressed most of the concerns raised by the previous review. Importantly, the authors clarified the novelty and significance of their findings. The authors claim that a major finding of their work is that higher visual areas (HVAs) are pivotal in shaping the response properties of individual primary cortex neurons in response to illusory brightness.

The work is significant and advances our understanding of contextual influence on the responses of primary visual cortex neurons. It is exciting that the authors identified a manipulation (silencing HVAs) that differentially impacts the response to the illusion and standard grating stimuli.

However, to fully support the claim noted above, it would be useful if the authors could perform additional analysis to demonstrate more precisely, what this pivotal role is, at the level of individual neurons. Specifically, (1) report the change in average response (e.g. Fig. 5f) on a cell by cell basis separately for each of the six animals [assuming that recordings were done with and without silencing in the same recording session]. (2) provide an estimation of the fraction of the HVA silenced during optogenetic stimulation and whether the HVA neurons that are silenced are retinotopically matched to the recorded V1 neurons - presumably, the entire HVA is not silenced or that would likely spill into V1, so providing information about what the manipulation is functionally doing within the HVAs is important.

We thank the Reviewer for their constructive feedback and for recognizing the significance of our work in understanding the contextual influence on the responses of primary visual cortex neurons. We are pleased that our manuscript has been well-received and appreciate their suggestions for further strengthening our findings.

Following the Reviewer's recommendation, we added a scatter plot showing each neuron's responses to the stimuli with and without opto-inhibition in V1 (Supplementary Fig. 14). This plot provides a detailed and precise visualization of the individual neuronal responses across all six animals, offering a more subtle understanding of how HVAs influence neurons in the primary visual cortex. We also confirm that recordings were done with and without silencing in the same recording session. We have clarified this in the methods section (see lines 753-755). We acknowledge the importance of estimating the fraction of the HVA silenced during optogenetic stimulation. In line with the Reviewer's suggestion, we report the fraction of inhibited cells in the HVA in the manuscript (lines 454-458). We also have shown the response of individual single units in HVAs to the LDG stimulus and LDG+light stimulus in Supplementary Fig. 13.

However, we note a potential limitation: In our experimental setup, we used a single shank laminar electrode, which was inserted at an oblique angle to reach several visual areas simultaneously, as illustrated in Figure 5a of our manuscript. While effective for accessing multiple visual areas, this approach presents a significant challenge in recording neurons with matching RFs. As depicted in Figures 5a and 5c, the RF locations vary along the horizontal axis across the probe, but there is also movement in the vertical location. This variability in RF positioning means that achieving a recording where the RFs of the HVA and V1 neurons precisely match is exceptionally challenging, if possible, given the constraints of our current methodology. We have discussed this limitation in our manuscript, emphasizing that while we

can infer the overall influence of HVAs on V1 responses, the precise retinotopic relationship between these areas remains an open question for future studies (line 623-628).

Reviewer #3 (Remarks to the Author):

The authors have performed substantial new experiments, analysis, and clarifications of text that address most of the major comments and substantially improve the manuscript. There are only a few remaining minor issues that can be addressed with text revisions.

We are grateful for the reviewer's positive feedback on the revisions and additional experiments conducted for our manuscript. We appreciate their recognition of our efforts to address their major comments from the previous review. Below, we respond to their minor comments point by point.

Reviewer 3, Minor comment #1

The addition of the pupil data is compelling. We would suggest condensing/moving some of the text in lines 473-501 to the introduction to motivate and highlight the finding for the reader earlier in the manuscript. Also, can the authors mention if they checked whether the overall luminance differences explain the timing /magnitude (not polarity) of the response?

We agree with the Reviewer's suggestion to move some text from lines 473-501 to the Introduction. This will help highlight the significance of the pupil data earlier in the manuscript, thereby better motivating the findings for the reader. We followed the reviewer's suggestion and revised the Introduction (lines 59-65).

Concerning the Reviewer's query about whether we checked if the overall luminance differences explain the response's timing or magnitude (but not polarity), we considered the impact of luminance differences on pupil response. In our analysis, we checked for overall luminance differences to ensure that the observed effects were attributable to the illusory stimuli rather than mere luminance changes. The luminance for NCS and DBC reduces by 0.82% during stimulus presentation compared to pre-stimulus luminance. However, the pupil responses to these two stimuli are different (Fig. 6). Conversely, luminance for the LDG stimulus increases by 23.35%. Although the luminance change in LDG, versus NCS, is 28 times higher, the max pupil response in these two conditions is similar (~8 % constriction in response to LDG and ~6% dilation in response to NCS). These results demonstrate that luminance differences do not explain the pupil response magnitude. This was an essential aspect of our methodology to isolate the specific effects of the illusion. We clarified this point in the manuscript, ensuring that the reader understands how we accounted for luminance differences in our analysis of the pupil data (lines 515-523).

Reviewer 3, Minor comment #2

The new data showing similar NCS response properties for units with RFs that do not overlap the inducer (non-overlapping) strengthens the claim that neurons in V1 are responding to the illusory brightness. However, some details of these experiments are missing from the Methods. Please make sure to include: Description of the new stimuli with wider spacing between inducers; RF mapping protocol (both "square" and "rectangle" mapping stimuli are referenced in the Results, but only squares are mentioned in the Methods). ; analytic methods for quantifying degree of overlap). It would also be great to include the RF centers and/or RF contours of several examples or the population average of all of the "non-overlapping" units on the same plot relative to the inducers (similar to 3a). This would provide a very convincing demonstration for the readers, and could provide a visual anchor for the authors' explanation

for the relationship to surround suppression that is unable to be addressed here (prior main comment #4).

We apologize for the discrepancy in the text. The stimuli used for the RF mapping session were the same for both experiments. As it was not a perfect square, we decided to refer to it as a rectangle (which is more precise). Accordingly, in the revised version of the manuscript, we changed the square to rectangle.

Moreover, we have followed the Reviewer's suggestion and revised the manuscript as follows:

A description of the new stimuli with broader spacing between inducers is presented in lines 726-731 of the Methods part. We added the description of the calculation of overlapping RF in the Methods (lines 822-832). We also, added a Supplementary Figure that displays examples of non-overlapping RFs. This figure includes RF centers and RF contours for several examples of "non-overlapping" units plotted relative to the inducers (Supplementary Fig. 16). We believe that these editions has enhanced the manuscript and thank the reviewer for their constructive suggestions.

Reviewer 3, Minor comment #3

The new experiments demonstrate a role for HVA inputs in specifically driving NCS response. However, the responses to NCS and LDG in LM and LI during the optogenetic manipulation are not shown. Please show a figure or report text results that quantify the responses to the NCS and LDG in LM and LI units during the optogenetic manipulation to demonstrate that they are adequately inhibited.

Following the reviewer's suggestion, we added Supplementary Fig. 13 to demonstrate the inhibition of HVAs during optogenetics manipulation. We also reported the fraction single units in LM and LI for which optogenetic manipulation altered the NCS-evoked response (Lines 454-458).

Reviewer 3, Minor comment #4

It is worth highlighting in text that the "rectangle latency" results along with laminar analysis demonstrate that the NCS responses do not cleanly map onto laminar or latency-based hierarchical organization. Perhaps worth speculating why (esp. in light of optogenetic results showing some sort of hierarchical influence).

In agreement with the Reviewer's suggestion, we highlighted this observation more prominently in the text. We now included text that emphasizes these findings and discusses some potential reasons for these findings (lines 631-638). We summarize these changes, as follows:

The hierarchical measures like laminar properties and rectangle latency do not fully conform to the expected patterns for illusion processing. On the other hand, the optogenetics findings provide compelling evidence that at least a portion of the illusion response is modulated by top-down modulation. Collectively, these results suggests a more complex interplay between bottom-up and top-down processing in visual perception than might be expected in a strictly hierarchical system. Therefore, traditional hierarchical models alone may not explain the processing of visual illusions.

Reviewer 3, Minor comment #5

The new data and analysis show that after controlling for physical luminance, interneurons and excitatory neurons have similar preference for NCS. However, in line 403 the authors suggest that the results demonstrate a role of interneurons in “spatial spreading and filling-in effect” which is not very well explained. How could inhibitory neurons (that are suppressing the transmission of excitatory signals beyond V1) accomplish this?

We appreciate the Reviewer's request for further clarification on the role of inhibitory interneurons (I units) in the spatial spreading and filling-in effect associated with the NCS illusion.

The analysis, mainly when focusing on non-overlapping units, revealed intriguing aspects of the response properties of inhibitory interneurons. We observed that for these units, the NCS delay index, which characterizes the response latency to the NCS stimulus relative to the rectangle latency, showed a significant positive correlation with the distance between the unit's receptive field (RF) and the center of the closest inducer (Pearson's correlation coefficient: $\rho=0.21$, $p=1.46e-3$). This finding suggests that the further the RF of an inhibitory neuron is from the inducer, the longer the latency of its response to the NCS stimulus.

This observation is particularly interesting as it implies a potential role for inhibitory interneurons in the mechanism underlying the filling-in effect of the NCS illusion. The increased latency with distance could indicate that these neurons are involved in progressively integrating or interpolating the visual information across the visual field, contributing to the representation of a continuous illusory grating. This process is essential for the brain to 'fill in' the gaps in visual information, especially in the context of illusions like the NCS, where the brain confronts ambiguous or incomplete visual cues.

However, at the level of circuit mechanisms, the role of inhibitory neurons in this process is not straightforward. How these neurons, given their inhibitory effects, contribute to the spatial spreading and filling-in effect is an intriguing question. The findings may indicate a nuanced role for inhibitory interneurons in visual processing, which involves network effects and other neural mechanisms beyond simple suppression.

In the revised manuscript, we have expanded our discussion to include these points, accordingly (lines 601-610).

Reviewer 3, Minor comment #6

The claim that complex cells contribute more to NCS processing is referenced several times (Line 379, 568) but the IGR shows no correlation with CSM index (Supp. Fig. 10), demonstrating that simple and complex cells have similar preferences for NCS vs LDG. IGR_F1 is correlated to CSM, but this does not mean that complex cells are more strongly involved in NCS processing than simple cells. For example, this effect could be due to the low levels of LDG F1 modulation rather than NCS F1 modulation. Therefore, the language regarding the preferential processing of NCS by complex cells (Line 379, 417, 568) should be tempered.

We are grateful for the Reviewer's constructive feedback, which has helped us improve the clarity of our manuscript. Following their suggestion, we have tempered the statements in our

manuscript regarding the preferential processing of NCS by complex cells. We have changed the language in the relevant sections (now lines 430-431, and 589-595) in the new version of the manuscript) to reflect the findings more accurately.